# On quantum separation of variables
# beyond fundamental representations

**Jean Michel Maillet[⋆] and Giuliano Niccoli[†]**

Univ Lyon, Ens de Lyon, Univ Claude Bernard Lyon 1, CNRS,
Laboratoire de Physique, UMR 5672, F-69342 Lyon, France

⋆ maillet@ens-lyon.fr, † giuliano.niccoli@ens-lyon.fr

## Abstract

We describe the extension, beyond fundamental representations of the Yang-Baxter algebra, of our new construction of separation of variables bases for quantum integrable lattice models. The key idea underlying our approach is to use the commuting conserved charges of the quantum integrable models to generate bases in which their spectral problem is *separated*, i.e. in which the wave functions are factorized in terms of specific solutions of a functional equation. For the so-called "non-fundamental" models we construct two different types of SoV bases. The first is given from the fundamental quantum Lax operator having isomorphic auxiliary and quantum spaces and that can be obtained by fusion of the original quantum Lax operator. The construction essentially follows the one we used previously for fundamental models and allows us to derive the simplicity and diagonalizability of the transfer matrix spectrum. Then, starting from the original quantum Lax operator and using the full tower of the fused transfer matrices, we introduce a second type of SoV bases for which the proof of the separation of the transfer matrix spectrum is naturally derived. We show that, under some special choice, this second type of SoV bases coincides with the one associated to Sklyanin's approach. Moreover, we derive the finite difference type (quantum spectral curve) functional equation and the set of its solutions defining the complete transfer matrix spectrum. This is explicitly implemented for the integrable quantum models associated to the higher spin representations of the general quasi-periodic $Y(gl_2)$ Yang-Baxter algebra. Our SoV approach also leads to the construction of a $Q$-operator in terms of the fused transfer matrices. Finally, we show that the $Q$-operator family can be equivalently used as the family of commuting conserved charges enabling to construct our SoV bases.


# 1  Introduction

In this article we continue the development of our approach [1–3] to generate the separation of variables (SoV) complete characterization of the spectrum of quantum integrable lattice models. We use the framework of the quantum inverse scattering method [4–12] and its associated Yang-Baxter algebra. Let us stress that in this context, the quantum version of the separation of variables has been pioneered by E. K. Sklyanin in a series of beautiful seminal works [13–18]. The main motivation to develop this new paradigm was to overcome several difficulties in applying the algebraic Bethe ansatz (ABA) in several important cases like the open Toda chain, in particular linked to the absence of an obvious reference state, see e.g. [13]. More conceptually, it was also designed to have a resolution scheme at the quantum level that would be the analog of the standard Hamilton-Jacobi method in classical Hamiltonian mechanics, see e.g. [19]. In particular, the main feature of the SoV method is that it is not an ansatz. As such, it leads to the possibility to find the complete characterization of the spectrum of quantum integrable models, this completeness question being in general a difficult task within the ABA method, see e.g. [20, 21].

The key ingredient of Sklyanin's approach is the construction, from the generators of the Yang-Baxter algebra, of two operator families $B(\lambda)$ and $A(\lambda)$, depending on a complex spectral parameter $\lambda \in \mathbf{C}$, and satisfying the following properties. The $B$-family must be a commuting family of simultaneously diagonalizable operators having simple spectrum. The separate variables are then given by the complete set of commuting operators $Y_n$ such that $B(Y_n) = 0$. Their common eigenbasis defines the SoV basis. The $A$-family forms also a continuous set of commuting operators, with simple spectrum, which, thanks to their commutation relations with the $B$-family stemming from the Yang-Baxter algebra, define the shift operators over the spectrum of the separate variables. Moreover, the $A$-family and the transfer matrices of the model satisfy over the spectrum of the separate variables closed characteristic equations (the analog of the Hamilton-Jacobi equations), the so-called quantum spectral curve equation that characterizes the spectrum of the given model.

This beautiful Sklyanin's picture for the construction of the SoV basis therefore requires

the proper identification of the operator families $B(\lambda)$ and $A(\lambda)$ and the proof that they indeed satisfy all the outlined required properties. Sklyanin has proposed how to construct these operator families for a large class of models associated to the representation of the 6-vertex Yang-Baxter algebra and even for the higher rank cases. Since then, this method has been successfully implemented, and in some cases partially generalized, to achieve the complete spectrum characterization of several classes of integrable quantum models mainly associated to different representations of the 6-vertex and 8-vertex Yang-Baxter algebras and reflection algebras as well as to their dynamical deformations [22–56]. Despite its many successes, this construction does not seem however to be completely universal; in particular, some difficulties arise already for the proper identification of the $A$ operator family for the fundamental representations of the higher rank Yang-Baxter algebras, see e.g. [1], although there was recently progresses to identify the proper $B$ operator, its spectrum, and how it can be used to obtain some eigenstates of the transfer matrix, see [57–59], however still insufficient to realize the full SoV program.

This motivated us to look for a different way for constructing the SoV basis that would not rely on finding such two families of operators $A$ and $B$. The new idea underlying our approach is to use the action of a well chosen set of commuting conserved charges on some generic co-vector to generate an Hilbert space basis in which their spectral problem is separated. In all the models we considered so far [1–3] this set is generated by the transfer matrix itself and provides effectively an SoV basis in which its spectrum can be characterize completely. In particular in such a basis the eigenvectors of the transfer matrix have coordinates given by the products of the corresponding transfer matrices eigenvalues. Hence, the resolution of the spectral problem determines not only the eigenvalues but also gives an algebraic construction of the corresponding eigenvectors in this SoV basis, which is a quite remarkable feature. It is to be emphasized that, in our approach, the SoV basis is directly generated by the quantum symmetries of the considered integrable model. Of course, such a program requires the proof of two main non-trivial steps. First, by using an appropriate set of commuting conserved charges we have to show that we can indeed construct a basis of the given space of the representation of the Yang-Baxter algebra. Second, we need to prove that their spectral problem is indeed separated in this basis. It means that all the wavefunctions should have a factorized form in terms of a well defined class of solutions of an appropriate functional equation (the quantum spectral curve equation). This amounts in fact to get the action of the transfer matrix in this basis which is itself given in terms of the transfer matrix action on the generating co-vector. It turns out that this is equivalent to identify the structure constants of the associative and commutative algebra of the conserved charges generated by the transfer matrices. Again, in all cases we have considered, these structure constants can be computed from the set of fusion relations satisfied by the transfer matrix and the associated quantum determinant evaluated in some specific points that in fact determines the separate variables.

In our previous articles [1–3], we have considered quantum integrable lattice models associated to fundamental representations of the Yang-Baxter algebra $Y(gl_n)$ and $U_q(gl_n)$ for arbitrary integer $n \geq 2$. In particular, in our first paper [1] we have presented how our new approach to construct the SoV bases for integrable quantum models associated to the fundamental representations of $Y(gl_n)$ with the most general quasi-periodic boundary conditions, and for some simple generalizations of them. Then, we have used it to obtain the explicit and complete characterization of the transfer matrix spectrum first for the cases of $Y(gl_2)$ and $U_q(sl_2)$ and then for $Y(gl_3)$. In our second paper these results on the complete transfer matrix spectrum have been extended to the case $Y(gl_n)$, for any integer $n \geq 2$ while in our third article, we have obtained similar results for the $U_q(gl_n)$ case. In our two first papers for the fundamental representations of $Y(gl_n)$, $n \geq 2$, we have identified a natural choice of the set of the commuting conserved charges and as well characterized the *generating co-vector* to be

used as starting point to generate our SoV basis. This has been motivated by the consequent simplicity of the proof that this system of co-vectors forms indeed a basis of the Hilbert space and that the spectrum of the transfer matrix is indeed separated in such a basis. The results are the introduction of the so-called quantum spectral curve and the exact characterization of the set of its solutions which generates the complete transfer matrix spectrum associating to any solution exactly one nonzero eigenvector up to trivial normalisation. These results allow also to point out how the SoV basis in our construction can be equivalently obtained by the action of the Baxter's $Q$-operator family [60–82] satisfying with the transfer matrices the quantum spectral curve equation. In our first paper [1] we have also shown that, under some specific choice of the co-vector, our SoV basis coincides with Sklyanin's SoV basis, when Sklyanin's approach applies, for integrable quantum models associated to rank one Yang-Baxter algebra. Based on our analysis of the $Y(gl_3)$ case, we also conjectured (and verified on small size chains) that the same should hold for the higher rank cases as well, the recent analysis [57–59] confirming such a statement for $Y(gl_n)$.

The aim of the present article is to show how our method works in cases going beyond the fundamental representations. Our interest in this situation, besides broadening the application of our method, is to understand and explain how our SoV construction works when we have at our disposal a richer structure of commuting conserved charges. As the current paper is mainly addressed to explain these features, we have chosen to consider the simplest example in this class, namely quantum integrable models associated to the higher spin representations of the rational 6-vertex Yang-Baxter algebra, i.e. $Y(gl_2)$. In doing so we also solve the case associated to the most general quasi-periodic boundary conditions[1]. Completely similar results can be derived for others compact non-fundamental representations, as for example the higher spin and cyclic representations of the trigonometric 6-vertex Yang-Baxter algebra or their higher rank cases as it will be described in future publications.

In this paper we first implement, for these higher-spin representations of the rational 6-vertex Yang-Baxter algebra, Sklyanin's construction for the SoV basis for the most general quasi-periodic integrable boundary conditions. This construction leads to generate the eigenbasis of the twisted $B$-family of commuting operators, or some simple generalization of it, and to prove that it is diagonalizable and simple spectrum for these higher spin representations.

Then, we show that our SoV construction presented in the fundamental representation in [1] can be indeed naturally extended to the compact non-fundamental representations. This is first done by substituting in the SoV basis construction the transfer matrix, i.e. the one associated to the trace over the two-dimensional auxiliary space, by the fundamental transfer matrix, i.e. the one associated to the trace over the auxiliary space isomorphic to the one of the local quantum spaces. We give the proof that the SoV basis construction can in this framework be derived following a method very similar to the one used for fundamental representations. One direct consequence of this SoV basis construction is then the simplicity and diagonalizability of the transfer matrix.

Then a second SoV basis construction is presented using the full tower of higher fused transfer matrices. This construction appears to be very natural as the action of the transfer matrix in this basis is easily computed just using the fusion relations. In particular, it allows to prove that the transfer matrix spectral problem is indeed separated in this basis. In fact, we then derive the quantum spectral curve equation and uniquely determine the set of its solutions that characterize the complete spectrum of the transfer matrices. We further show that our second SoV basis indeed coincides with Sklyanin's one once we chose the generating co-vector in an appropriate way.

Finally, we show that the quantum spectral curve equation together with the diagonaliz-

---

[1]Indeed, only the case associated to the anti-periodic boundary conditions was solved previously in [44].

ability and the simple spectrum character of the transfer matrix family allow us to characterize the Q-operator family in terms of the elements of the monodromy matrix, and in particular in terms of the set of fused transfer matrices themselves. This result also allows us to rewrite our second SoV basis as the action of the Q-operator family on some new generating co-vector, i.e. to use the Q-operator family as the set of commuting conserved charges generating our SoV basis. The striking effect in using the basis generated by the Q-operator is that the action of the transfer matrix on this SoV basis is directly given as an explicit linear action thanks to the T-Q equation. Hence it realizes the key feature of the the Frobenius method described in [1], here generalized to the transfer matrix, to have a basis for which the linear action of the transfer matrix on it is just given by the characteristic equation (here the T-Q equation) determining its spectrum.

It is worth to comment that on the basis of all our current results for both fundamental and non-fundamental compact representations of the Yang-Baxter algebra our SoV construction based on the use of the Q-operator family as the generating set of commuting conserved charges always lead to the same natural choice of the SoV basis induced by the fusion of transfer matrices.

Clearly in order to use directly the Q-operator family to generate SoV bases for others integrable quantum lattice models all the following fundamental elements have to be accessible: first we need to have an SoV independent characterization of the Q-operator family; second we have to design some criteria to identify appropriate generating co-vectors (as starting point of our SoV construction) as well as the exact subset of commuting conserved charges in the Q-operator family (i.e. the spectrum of the separate variables); third a proof that the set of co-vectors generated is indeed a basis; fourth that the transfer matrix spectrum is indeed separated in this basis.

In fact, it is important to stress that in our current construction it is indeed the structure of the transfer matrix fusion relations and the fact that they simplify for special choices of the spectral parameters that allows us to naturally select the subset of commuting conserved charges to be used to generate the SoV basis. Furthermore, these fusion relations determine the structure constants of the commutative (associative) algebra of conserved charges generated by the transfer matrices.

The present paper is organized in five sections. In section 2, we recall the higher spin representations of the rational rank one Yang-Baxter algebra and the fused transfer matrix general properties. In section 3, we present Sklyanin's type SoV basis construction giving an explicit representation of its co-vectors. In section 4, we present our SoV basis construction and the consequent complete characterization of the transfer matrix spectrum. In subsection 4.1, this is done producing an SoV basis which is the natural generalization of those generated in the case of the fundamental representations in [1–3]. In subsection 4.2, we present a second SoV basis on which the action of the transfer matrix is easily computed by using the fusion relations. This new basis is shown there to coincide with Sklyanin's one under a proper choice of the generating co-vector. Finally, in subsection 5.1 we prove the reformulation of the discrete SoV complete spectrum characterization in terms the so-called quantum spectral curve equation. This last result allows us to determine the Q-operator in subsection 5.2 while we use it to reconstruct our second SoV basis in subsection 5.3. Finally, in the Conclusion we discuss, on general ground, the relations between the different SoV bases presented in this paper.

# 2 The quasi-periodic $Y(gl_2)$ higher spin representations

Let us recall that the first studies of the integrable higher spin quantum Heisenberg chains have been developed in [83–95]. The next two subsections are used to recall the higher spin representations of the rank one rational Yang-Baxter algebra and the properties of the fused transfer matrices which will be used in the next sections to develop our analysis in the framework of the separation of variables.

## 2.1 Higher spin representations

The generators of the $sl(2)$ algebra:

$$[S^z, S^\pm] = \pm S^\pm, \quad [S^+, S^-] = 2S^z, \tag{2.1}$$

admit the following spin-$s_n$ representation:

$$S_n^z = \mathrm{diag}(s_n, s_n - 1, \ldots, -s_n), \quad S_n^+ = \left(S_n^-\right)^t = \begin{pmatrix} 0 & x_n(1) & & \\ & \ddots & \ddots & \\ & & \ddots & x_n(2s_n) \\ & & & 0 \end{pmatrix}, \tag{2.2}$$

where $x_n(j) \equiv \sqrt{j(2s_n + 1 - j)}$, in a spin-$s_n$ representation associated the linear space $V^{(2s_n)} \simeq \mathbb{C}^{2s_n+1}$ with $2s_n \in \mathbb{Z}^{>0}$. Then, the following Lax operator:

$$\mathsf{L}_{0n}^{(1,2s_n)}(\lambda) \equiv \begin{pmatrix} \lambda + \eta(1/2 + S_n^z) & \eta S_n^- \\ \eta S_n^+ & \lambda + \eta(1/2 - S_n^z) \end{pmatrix}_{[0]} \in \mathrm{End}(V_0^{(1)} \otimes V_n^{(2s_n)}), \tag{2.3}$$

associated to each local quantum space $V_n^{(2s_n)}$, satisfies the following Yang-Baxter algebra:

$$R_{12}(\lambda - \mu)\mathsf{L}_{1n}^{(1,2s_n)}(\lambda)\mathsf{L}_{2n}^{(1,2s_n)}(\mu) = \mathsf{L}_{2n}^{(1,2s_n)}(\mu)\mathsf{L}_{1n}^{(1,2s_n)}(\lambda)R_{12}(\lambda - \mu), \tag{2.4}$$

associated to the rational 6-vertex $R$-matrix:

$$\mathsf{L}_{ab}^{(1,1)}(\lambda) \equiv R_{ab}(\lambda) \equiv \begin{pmatrix} \lambda + \eta & 0 & 0 & 0 \\ 0 & \lambda & \eta & 0 \\ 0 & \eta & \lambda & 0 \\ 0 & 0 & 0 & \lambda + \eta \end{pmatrix} \in \mathrm{End}(V_a^{(1)} \otimes V_b^{(1)}). \tag{2.5}$$

The scalar Yang-Baxter equation:

$$R_{ab}(\lambda)K_a K_b = K_b K_a R_{ab}(\lambda), \tag{2.6}$$

is satisfied by any $K \in \mathrm{End}(\mathbb{C}^2)$, which defines the $gl_2$ invariance of the rational 6-vertex $R$-matrix. We can then introduce the monodromy matrix associated to a quantum lattice model with N sites, twist $K$, and carrying at each site $n \in \{1, \ldots, N\}$ a representation $V_n^{(2s_n)}$:

$$\begin{aligned} \mathsf{M}_0^{(K|1)}(\lambda) &\equiv \begin{pmatrix} A^{(K)}(\lambda) & B^{(K)}(\lambda) \\ C^{(K)}(\lambda) & D^{(K)}(\lambda) \end{pmatrix}_{[0]} \\ &\equiv K_0 \mathsf{L}_{0N}^{(1,2s_N)}(\lambda - \xi_N) \cdots \mathsf{L}_{01}^{(1,2s_1)}(\lambda - \xi_1) \in \mathrm{End}(V_0^{(1)} \otimes \mathcal{H}), \end{aligned} \tag{2.7}$$

where we have defined $\mathcal{H} = \otimes_{n=1}^{N} V_n^{(2s_n)}$ and the $\xi_n$ are the inhomogeneity parameters. In the following we will assume these parameters to be in generic positions, namely $\xi_i \neq \xi_j (\mathrm{mod}\,\eta)$

whenever $i \neq j$. In the following, we will denote by $k_1$ and $k_2$ the eigenvalues of such $2 \times 2$ twist matrix $K$ that we assume to be distinct and non zero (more general cases could be considered however). This monodromy matrix satisfies also the same rational 6-vertex Yang-Baxter algebra:

$$R_{12}(\lambda - \mu)M_1^{(K|1)}(\lambda)M_2^{(K|1)}(\mu) = M_2^{(K|1)}(\mu)M_1^{(K|1)}(\lambda)R_{12}(\lambda - \mu) \in \mathrm{End}(V_a^{(1)} \otimes V_b^{(1)} \otimes \mathcal{H}), \quad (2.8)$$

which implies that the transfer matrix:

$$\mathsf{T}^{(K|1)}(\lambda) = \mathrm{tr}_0[\mathsf{M}_0^{(K|1)}(\lambda)], \quad (2.9)$$

is a one-parameter family of commuting operators and that the quantum determinant:

$$\Delta_\eta^{(K)}(\lambda) \equiv \mathrm{qdet}\mathsf{M}_0^{(K|1)}(\lambda) \equiv \mathsf{A}^{(K)}(\lambda)\mathsf{D}^{(K)}(\lambda - \eta) - \mathsf{B}^{(K)}(\lambda)\mathsf{C}^{(K)}(\lambda - \eta) \quad (2.10)$$

is a central element of the Yang-Baxter algebra of the following form:

$$\Delta_\eta^{(K)}(\lambda) = \mathrm{qdet}\mathsf{M}_0^{(K|1)}(\lambda) \equiv \det K \, \mathrm{qdet}\mathsf{M}_0^{(I|1)}(\lambda), \quad \mathrm{qdet}\mathsf{M}_0^{(I|1)}(\lambda) = a(\lambda)d(\lambda - \eta), \quad (2.11)$$

where $\mathsf{M}_0^{(I|1)}(\lambda)$ is the monodromy matrix associated to the $2 \times 2$ identity twist matrix $K = \mathbb{I}_{2 \times 2}$, and

$$a(\lambda) = \prod_{n=1}^{\mathsf{N}}\left(\lambda - \xi_n^- + s_n\eta\right), \qquad d(\lambda) = \prod_{n=1}^{\mathsf{N}}\left(\lambda - \xi_n^- - s_n\eta\right), \quad (2.12)$$

and we have used the notation $\lambda^\pm \equiv \lambda \pm \eta/2$. With these notations the quantum determinant is given by $\Delta_\eta^{(K)}(\lambda) = \mathsf{k}_1\mathsf{k}_2 a(\lambda)d(\lambda - \eta)$. Moreover we will use the following shorthand notations for the shifted inhomogeneities:

$$\xi_n^{(k_n)} \equiv \xi_n^- + (s_n - k_n)\eta, \quad (2.13)$$

with $h_n \in \{0, ..., 2s_n\}$ for all the $n \in \{1, ..., \mathsf{N}\}$. Hence, we get:

$$a(\lambda) = \prod_{n=1}^{\mathsf{N}}\left(\lambda - \xi_n^{(2s_n)}\right) \text{ and } d(\lambda) = \prod_{n=1}^{\mathsf{N}}\left(\lambda - \xi_n^{(0)}\right). \quad (2.14)$$

## 2.2 Fusion relations for higher spin transfer matrices

The fusion procedure was first developed in [84] for the case of the rational 6-vertex representations of the type analyzed here and later in [92] for the trigonometric ones.

Let us define the following symmetric and antisymmetric projectors:

$$P_{1...m}^\pm = \frac{1}{m!}\sum_{\pi \in S_m}(\pm 1)^{\sigma_\pi}P_\pi, \quad (2.15)$$

where $P_\pi$ is the permutation operator:

$$P_\pi(v_1 \otimes \cdots \otimes v_m) = v_{\pi(1)} \otimes \cdots \otimes v_{\pi(m)}, \quad \forall v_1 \otimes \cdots \otimes v_m \in \otimes_{a=1}^m V_a, \quad (2.16)$$

with $P_1^- = I$. Note that in our current representations, we have that $V_a \simeq \mathbb{C}^2$ and the $P_{1...m}^+$ is a rank $m + 1$ projector, so that,

$$V_{1...m}^+ \equiv P_{1...m}^+(\otimes_{a=1}^m V_a) \simeq \mathbb{C}^{m+1} \quad (2.17)$$

is an $(m+1)$-dimensional vector space. Then, we can define the following higher transfer matrices:

$$\mathsf{T}^{(K|a)}(\lambda) \equiv \mathrm{tr}_{V_{1\dots a}^+} \mathsf{M}_{1\dots a}^{(K|a)}(\lambda) \in \mathrm{End}(\mathcal{H}) \quad \forall \lambda \in \mathbb{C}, a \in \mathbb{Z}^{>0}, \tag{2.18}$$

where we have defined the higher spin monodromy matrices by:

$$\mathsf{M}_{1\dots a}^{(K|a)}(\lambda) \equiv P_{1\dots a}^+ \mathsf{M}_1^{(K|1)}(\lambda+(a-1)\eta)\cdots \mathsf{M}_{a-1}^{(K|1)}(\lambda+\eta)\mathsf{M}_a^{(K|1)}(\lambda)P_{1\dots a}^+ \in \mathrm{End}(V_{1\dots a}^+ \otimes \mathcal{H}), \tag{2.19}$$

for which the following identity holds:

$$\mathsf{M}_{1\dots a}^{(K|a)}(\lambda) = K_{1\dots a}^{(a)} \mathsf{M}_{1,\dots,a}^{(I|a)}(\lambda), \tag{2.20}$$

with

$$K_{1\dots a}^{(a)} \equiv P_{1\dots a}^+ K_1 \cdots K_a P_{1\dots a}^+ \in \mathrm{End}(V_{1\dots a}^+). \tag{2.21}$$

These transfer matrices define commuting families of operators satisfying for any values of $\lambda$ and $\mu$ and positive integers $l,m \in \mathbb{N}^*$:

$$[\mathsf{T}^{(K|l)}(\lambda), \mathsf{T}^{(K|m)}(\mu)] = 0 \quad, \tag{2.22}$$

satisfying the fusion relations:

$$\mathsf{T}^{(K|l+1)}(\lambda) = \mathsf{T}^{(K)}(\lambda+l\eta)\mathsf{T}^{(K|l)}(\lambda) - \Delta_\eta^{(K)}(\lambda+l\eta)\mathsf{T}^{(K|l-1)}(\lambda), \tag{2.23}$$

where for simplicity we used the following notations:

$$\mathsf{T}^{(K)}(\lambda) \equiv \mathsf{T}^{(K|1)}(\lambda) \quad \text{and} \quad \mathsf{T}^{(K|0)}(\lambda) \equiv 1. \tag{2.24}$$

Furthermore, we also set $\mathsf{T}^{(K|l)}(\lambda) \equiv 0$ for any $l < 0$. These fusion relations define uniquely any higher spin transfer matrix $\mathsf{T}^{(K|l)}(\lambda)$ in terms of the transfer matrix $\mathsf{T}^{(K)}(\lambda)$. Moreover, it is possible to write an explicit determinant formula that solves the hierarchy of fusion relations as follows.

**Proposition 2.1.** *Let $D_l(\mathsf{T}^{(K)}(\lambda))$ be the following tridiagonal $l \times l$ matrix:*

$$D_l(\mathsf{T}^{(K)}(\lambda)) \equiv$$

$$\begin{pmatrix} \mathsf{T}^{(K)}(\lambda+(l-1)\eta) & -k_1 a(\lambda+(l-1)\eta) & 0\dots & & 0 \\ -k_2 d(\lambda+(l-2)\eta) & \mathsf{T}^{(K)}(\lambda+(l-2)\eta) & -k_1 a(\lambda+(l-2)\eta) & 0\dots & \vdots \\ 0\dots & \ddots & \ddots & \ddots & 0 \\ \vdots & \dots 0 & -k_2 d(\lambda+\eta) & \mathsf{T}^{(K)}(\lambda+\eta) & -k_1 a(\lambda+\eta) \\ 0\dots & \dots & \dots 0 & -k_2 d(\lambda) & \mathsf{T}^{(K)}(\lambda) \end{pmatrix}, \tag{2.25}$$

*then:*

$$\mathsf{T}^{(K|l)}(\lambda) = \det_l D_l(\mathsf{T}^{(K)}(\lambda)). \tag{2.26}$$

*Proof.* The proof can be done by an elementary induction. Indeed the formula trivially holds for $l = 1$ and for $l = 2$ it reduces to the fusion relation defining $\mathsf{T}^{(K|2)}(\lambda)$:

$$\begin{aligned} \mathsf{T}^{(K|2)}(\lambda) &= \mathsf{T}^{(K)}(\lambda+\eta)\mathsf{T}^{(K)}(\lambda) - \Delta_\eta^{(K)}(\lambda+\eta)\mathsf{T}^{(K|0)}(\lambda) \tag{2.27} \\ &= \mathsf{T}^{(K)}(\lambda+\eta)\mathsf{T}^{(K)}(\lambda) - k_1 k_2 a(\lambda+\eta)d(\lambda)\mathsf{T}^{(K|0)}(\lambda) \\ &= \det_2 D_2(\mathsf{T}^{(K)}(\lambda)). \end{aligned}$$

Let us now suppose the formula is true up to some integer $l \geq 2$. Then making the expansion of $\det_{l+1} D_{l+1}(\mathsf{T}^{(K)}(\lambda))$ with respect to its first column we get:

$$\det_{l+1} D_{l+1}(\mathsf{T}^{(K)}(\lambda)) = \mathsf{T}^{(K)}(\lambda + l\eta)\det_l D_l(\mathsf{T}^{(K)}(\lambda)) + \mathsf{k}_2 d(\lambda + (l-1)\eta)\det_l \Gamma_l(\lambda), \quad (2.28)$$

where the $l \times l$-matrix $\Gamma_l(\lambda)$ is given by:

$$\Gamma_l(\lambda) \equiv \begin{pmatrix} \begin{array}{c|c} -\mathsf{k}_1 a(\lambda + l\eta) & 0 \; \dots \\ \hline -\mathsf{k}_2 d(\lambda + (l-2)\eta) & \\ 0 & D_{l-1}(\mathsf{T}^{(K)}(\lambda)) \\ \vdots & \end{array} \end{pmatrix}. \quad (2.29)$$

Then expanding $\det_l \Gamma_l(\lambda)$ by its first row and applying the induction hypothesis for $l$ and $l-1$ we get:

$$\det_{l+1} D_{l+1}(\mathsf{T}^{(K)}(\lambda)) = \mathsf{T}^{(K)}(\lambda + l\eta)\mathsf{T}^{(K|l)}(\lambda) - \mathsf{k}_1 \mathsf{k}_2 a(\lambda + l\eta)d(\lambda + (l-1)\eta)\mathsf{T}^{(K|l-1)}(\lambda), \quad (2.30)$$

hence,

$$\begin{aligned}
\det_{l+1} D_{l+1}(\mathsf{T}^{(K)}(\lambda)) &= \mathsf{T}^{(K)}(\lambda + l\eta)\mathsf{T}^{(K|l)}(\lambda) - \Delta_\eta^{(K)}(\lambda + l\eta)\mathsf{T}^{(K|l-1)}(\lambda) \\
&= \mathsf{T}^{(K|l+1)}(\lambda), \quad (2.31)
\end{aligned}$$

which completes the proof. $\qquad \square$

From this determinant representation of the fused transfer matrices it is easy to derive another fusion relation by expanding now the corresponding determinant by its last column instead of its first one. Repeating the above steps we get:

$$\mathsf{T}^{(K|l+1)}(\lambda) = \mathsf{T}^{(K)}(\lambda)\mathsf{T}^{(K|l)}(\lambda + \eta) - \Delta_\eta^{(K)}(\lambda + \eta)\mathsf{T}^{(K|l-1)}(\lambda + 2\eta). \quad (2.32)$$

In fact, these two fusion relations (2.23) and (2.32) are two particular cases of more general fusion relations that can be derived thanks to the determinant of a tridiagonal matrix form of the level $l$ fused transfer matrix $\mathsf{T}^{(K|l)}(\lambda)$ given by (2.26).

**Proposition 2.2.** *For any $l \geq 0$ and any $j$ such that $0 \leq j \leq l$, the fused transfer matrix $\mathsf{T}^{(K|l)}(\lambda)$ has the following decomposition:*

$$\begin{aligned}
\mathsf{T}^{(K|l)}(\lambda) = {}& \mathsf{T}^{(K|j)}(\lambda + (l-j)\eta)\mathsf{T}^{(K|l-j)}(\lambda) \\
&- \Delta_\eta^{(K)}(\lambda + (l-j)\eta)\mathsf{T}^{(K|j-1)}(\lambda + (l-j+1)\eta)\mathsf{T}^{(K|l-j-1)}(\lambda). \quad (2.33)
\end{aligned}$$

*Proof.* The relation follows from the general relation (A.3) for the determinant of a tridiagonal matrix in terms of its sub-determinants that we describe in Appendix A. Applying this relation to (2.26), we get:

$$\begin{aligned}
\mathsf{T}^{(K|l)}(\lambda) = {}& \mathsf{T}^{(K|1)}(\lambda + (l-j)\eta)\mathsf{T}^{(K|j-1)}(\lambda + (l-j+1)\eta)\mathsf{T}^{(K|l-j)}(\lambda) \\
&- \Delta_\eta^{(K)}(\lambda + (l-j+1)\eta)\mathsf{T}^{(K|j-2)}(\lambda + (l-j+2)\eta)\mathsf{T}^{(K|l-j)}(\lambda) \\
&- \Delta_\eta^{(K)}(\lambda + (l-j)\eta)\mathsf{T}^{(K|j-1)}(\lambda + (l-j+1)\eta)\mathsf{T}^{(K|l-j-1)}(\lambda). \quad (2.34)
\end{aligned}$$

Then, using the elementary fusion relation (2.32) for $\mathsf{T}^{(K|j)}(\lambda + (l-j)\eta)$:

$$\begin{aligned}
\mathsf{T}^{(K|1)}(\lambda + (l-j)\eta)\mathsf{T}^{(K|j-1)}(\lambda + (l-j+1)\eta) = {}& \mathsf{T}^{(K|j)}(\lambda + (l-j)\eta) \\
&+ \Delta_\eta^{(K)}(\lambda + (l-j+1)\eta)\mathsf{T}^{(K|j-2)}(\lambda + (l-j+2)\eta) \\
&\qquad\qquad\qquad\qquad\qquad (2.35)
\end{aligned}$$

we can get rid of the term containing $\mathsf{T}^{(K|j-2)}(\lambda + (l-j+2)\eta)$, and obtain the desired result. Note that (2.33) can also be proven by direct induction. $\qquad \square$

In the characterization of the transfer matrix spectrum we will use the following property:

**Proposition 2.3.** *The transfer matrix* $\mathsf{T}^{(K)}(\lambda)$ *is a degree* $\mathsf{N}$ *polynomial in* $\lambda$ *with central asymptotics:*

$$\lim_{\lambda \to \infty} \lambda^{-\mathsf{N}} \mathsf{T}^{(K)}(\lambda) = tr_0 K_0, \tag{2.36}$$

*and it satisfies the following system of* $\mathsf{N}$ *equations:*

$$\mathsf{T}^{(K|2s_n+1)}(\xi_n^{(2s_n)}) = \det_{2s_n+1} D_{2s_n+1}(\mathsf{T}^{(K)}(\xi_n^{(2s_n)})) = 0, \quad \forall n \in \{1, ..., \mathsf{N}\}, \tag{2.37}$$

*where,* $D_{2s_n+1}(\mathsf{T}^{(K)}(\lambda))$ *is the* $(2s_n + 1) \times (2s_n + 1)$ *tridiagonal matrix defined in* (2.25).

*Proof.* The computation of the asymptotics is easily derived from the known asymptotics of the elements of the $R$-matrix $R_{an}^{(1|2s_n)}(\lambda) \in \mathrm{End}(V_a^{(1)} \otimes V_b^{(2s_n)})$. Moreover, it can be shown from [84] that the polynomials $\mathsf{T}^{(K|2s_n+1)}(\lambda)$ admit the central zeroes $\xi_n^{(2s_n)}$. Then the system of equations satisfied by the transfer matrix is just a direct consequence of the fusion relations for the transfer matrices resolved in (2.26) and of the emerging central zeroes above discussed. $\square$

For definiteness we will call "fundamental" the transfer matrices $\mathsf{T}^{(K|2s_a)}$, namely whenever the fusion index is such that on the site $a$ the quantum and auxiliary spaces are isomorphic. In particular, in the case where all values of spins $s_a$ are equal to some chosen $s$, such a fundamental transfer matrix is obtained from the product of fundamental $R$-matrices $R^{(2s|2s)}$.

Notice that, starting from the fact that $\xi_n^{(2s_n)}$ is a central zero of $\mathsf{T}^{(K|2s_n+1)}(\lambda)$, an elementary recursion, with the help of the fusion relations (2.23) and (2.32), can be used to prove that indeed the fused transfer matrix $\mathsf{T}^{(K|2s_n+l)}(\lambda)$, for $l \geq 1$ admits the set of central zeroes $\xi_n^{(2s_n)}, ..., \xi_n^{(2s_n+l-1)}$. This property also follows directly from representation theory for higher spin $R$-matrices [84]. The existence of these central zeroes identities implies some interesting particular fusion relations that we will use in our construction of the SoV bases. We summarize them in the following proposition.

**Proposition 2.4.** *For any* $a = 1, ..., N$ *and any* $j = 0, 1, ...., 2s_a$; *we have the following higher fusion relations in the well chosen shifted inhomogeneity points:*

$$\mathsf{T}^{(K|2s_a)}(\xi_a^{(2s_a)}) \mathsf{T}^{(K|j)}(\xi_a^{(j-1)}) = \mathsf{T}^{(K|2s_a-j)}(\xi_a^{(2s_a)}) \prod_{k=0}^{j-1} \Delta_\eta^{(K)}(\xi_a^{(k)}). \tag{2.38}$$

*Proof.* The relation is obviously true for $j = 0$ and reduces to the standard fusion relation for $j = 1$ when evaluated in $\lambda = \xi_a^{(2s_a)}$ using the fact that $\mathsf{T}^{(K|2s_a+1)}(\xi_a^{(2s_a)}) = 0$ due to the above central zero property as $\xi_a^{(2s_a)} = \xi_a^- - s_a \eta$. Then the relation can be easily proven by induction on $j$ using again the standard fusion relation evaluated in the shifted inhomogeneities. Note that we can also prove this relation by applying recursively (2.33) for $l = 2s_a + 1$ and varying values of $j$ at $\lambda = \xi_a^{(2s_a)}$. $\square$

Let us finally comment that if the original $2 \times 2$ twist matrix $K$ is diagonalizable, simple and invertible then the same is true for all the fused twist matrices. In particular, given the distinct nonzero eigenvalues $\mathsf{k}_1$ and $\mathsf{k}_2$ of such $2 \times 2$ twist matrix $K$ then the fused twist matrix $K^{(a)}$ has the following simple spectrum $\mathsf{k}_h = \mathsf{k}_1^{a+1-h} \mathsf{k}_2^{h-1}$ for all $h \in \{1, ..., a+1\}$, for any fixed $a \in \mathbb{N}^*$. Note that here and in the following we use a shorter notation to represent the spaces in the fused twist matrices, i.e. we can write $K_n^{(2s_n)}$ thanks to the isomorphism $V_n^{(2s_n)} \simeq V_{1,...,2s_n}^+$. Finally, let us remark that the following commutation relations hold:

$$[\mathsf{L}_{0n}^{(1,2s_n)}(\lambda), K_0 \otimes K_n^{(2s_n)}] = 0 \in \mathrm{End}(V_0 \otimes V_n^{(2s_n)}), \quad \forall a \in \{1, ..., \mathsf{N}\} \tag{2.39}$$

and so we have also:

$$[\mathsf{M}_0^{(I)}(\lambda), K_0 \otimes \mathsf{K}] = 0 \in \text{End}(V_0 \otimes \mathcal{H}), \quad \forall a \in \{1, ..., \mathsf{N}\}, \tag{2.40}$$

where:

$$\mathsf{K} \equiv \otimes_{n=1}^{\mathsf{N}} K_n^{(2s_n)} \in \text{End}(\mathcal{H}). \tag{2.41}$$

## 3 Sklyanin's type construction of the SoV basis

Let us first show how Sklyanin's approach to separation of variables (SoV) [13–18] works for the $\mathsf{T}^{(K)}$-spectral problem in representations for which the commutative family of operators $\mathsf{B}^{(K)}(\lambda)$ (or $\mathsf{C}^{(K)}(\lambda)$) is diagonalizable and with simple spectrum. As already explained in our previous paper for the case of fundamental representation such a statement can be extended by using the $gl_2$ invariance. In order to do so we have to use the following remark that given a

$$K = \begin{pmatrix} a & b \\ c & d \end{pmatrix} \neq \alpha \mathbb{I}_{2\times 2} \in \text{End}(\mathbb{C}^2), \tag{3.1}$$

either it satisfies the condition $b \neq 0$ (or $c \neq 0$) directly or there exists a $W^{(K)} \in \text{End}(\mathbb{C}^2)$ such that:

$$\bar{K} = \left(W^{(K)}\right)^{-1} K W^{(K)} = \begin{pmatrix} \bar{a} & \bar{b} \neq 0 \\ \bar{c} \neq 0 & \bar{d} \end{pmatrix}, \tag{3.2}$$

so that we can state the following:

**Theorem 3.1.** *If the inhomogeneities* $\{\xi_1, ..., \xi_{\mathsf{N}}\} \in \mathbb{C}^{\mathsf{N}}$ *satisfy the conditions:*

$$\xi_a \neq \xi_b \mod \eta \quad \forall a \neq b \in \{1, ..., \mathsf{N}\}, \tag{3.3}$$

*and the twist matrix*

$$K \neq \alpha \mathbb{I}_{2\times 2} \in \text{End}(\mathbb{C}^2), \tag{3.4}$$

*for any* $\alpha \in \mathbb{C}$, *i.e.* $K$ *is not proportional to the identity, then the* $\mathsf{T}^{(K)}$*-spectral problem admits Sklyanin's like separate variable representations. More in detail:*

*a) If* $b \neq 0$ *(or* $c \neq 0$*) then* $\mathsf{B}^{(K)}(\lambda)$ *(or* $\mathsf{C}^{(K)}(\lambda)$*) is diagonalizable and with simple spectrum and the quantum separate variables are generated by the* $\mathsf{B}^{(K)}$*-operator (or* $\mathsf{C}^{(K)}$*-operator) zeroes.*

*b) If* $b = 0$, *defined:*

$$\mathcal{W}_K \equiv \otimes_{n=1}^{\mathsf{N}} W_n^{(K)} \in \text{End}(\mathcal{H}), \tag{3.5}$$

*then*

$$\tilde{\mathsf{B}}^{(K)}(\lambda) = tr_{V_a}[W_a^{(K)} \begin{pmatrix} 0 & 0 \\ 1 & 0 \end{pmatrix}_a \left(W^{(K)}\right)_a^{-1} \mathsf{M}_a^{(K)}(\lambda)]$$
$$= \mathcal{W}_K \mathsf{B}^{(\bar{K})}(\lambda) \mathcal{W}_K^{-1}, \tag{3.6}$$

*is diagonalizable and with simple spectrum and the quantum separate variables are generated by the* $\tilde{\mathsf{B}}^{(K)}$*-operator zeroes.*

*c) If* $c = 0$ *then*

$$\tilde{\mathsf{C}}^{(K)}(\lambda) = tr_{V_a}[W_a^{(K)} \begin{pmatrix} 0 & 1 \\ 0 & 0 \end{pmatrix}_a \left(W^{(K)}\right)_a^1 \mathsf{M}_a^{(K)}(\lambda)]$$
$$= \mathcal{W}_K \mathsf{C}^{(\bar{K})}(\lambda) \mathcal{W}_K^{-1}, \tag{3.7}$$

*is diagonalizable and with simple spectrum and the quantum separate variables are generated by the* $\tilde{\mathsf{C}}^{(K)}$*-operator zeroes.*

In the next subsection we construct explicitly the $B^{(K)}$-eigenbasis and $\tilde{B}^{(K)}$-eigenbasis, the construction for $C^{(K)}$-eigenbasis and $\tilde{C}^{(K)}$-eigenbasis can be similarly derived, in this way giving a constructive proof of the above theorem.

## 3.1 Construction of the SoV representation in $B^{(K)}$-eigenbasis

Let

$$\langle 0| \equiv \otimes_{n=1}^{N} \langle 0, n|, \tag{3.8}$$

where we have defined for any value of $n$ the following $2s_n + 1$-co-vectors (*local left references states*):

$$\langle 0, n| = (1, 0, ..., 0)_{2s_n+1}, \tag{3.9}$$

and let us denote by $k_1$ and $k_2$ the eigenvalues of $K \in \text{End}(\mathbb{C}^2)$, then:

**Theorem 3.2.** *a) If the conditions in (3.3) are verified and $K \in \text{End}(\mathbb{C}^2)$ is any $2 \times 2$ matrix not proportional to the identity, invertible[2] and satisfying the condition $b \neq 0$, then the set of co-vectors $\langle \boldsymbol{h}|_{Sk} \equiv \langle h_1, ..., h_N|_{Sk}$, defined by:*

$$\langle \boldsymbol{h}|_{Sk} \equiv \langle 0| \prod_{n=1}^{N} \prod_{k_n=0}^{h_n-1} \frac{A^{(K)}(\xi_n^{(k_n)})}{k_1 a(\xi_n^{(k_n)})}, \tag{3.10}$$

*where $h_n \in \{0, ..., 2s_n\}$ for all the $n \in \{1, ..., N\}$ and:*

$$\xi_n^{(k_n)} \equiv \xi_n^- + (s_n - k_n)\eta, \tag{3.11}$$

*defines a co-vector $B^{(K)}$-eigenbasis of $\mathcal{H}$:*

$$\langle \boldsymbol{h}|_{Sk} B^{(K)}(\lambda) = d_{\boldsymbol{h}}^{(K)}(\lambda) \langle \boldsymbol{h}|_{Sk}, \tag{3.12}$$

*with the distinct eigenvalues:*

$$d_{\boldsymbol{h}}^{(K)}(\lambda) \equiv b \prod_{n=1}^{N} (\lambda - \xi_n^{(h_n)}) \quad and \quad \boldsymbol{h} \equiv (h_1, ..., h_N). \tag{3.13}$$

*Moreover it holds:*

$$\langle \boldsymbol{h}|_{Sk} A^{(K)}(\lambda) = \sum_{a=1}^{N} \prod_{b \neq a} \frac{\lambda - \xi_b^{(h_b)}}{\xi_a^{(h_a)} - \xi_b^{(h_b)}} k_1 a(\xi_a^{(h_a)}) \langle \boldsymbol{h}|_{Sk} T_a^+, \tag{3.14}$$

$$\langle \boldsymbol{h}|_{Sk} D^{(K)}(\lambda) = \sum_{a=1}^{N} \prod_{b \neq a} \frac{\lambda - \xi_b^{(h_b)}}{\xi_a^{(h_a)} - \xi_b^{(h_b)}} k_2 d(\xi_a^{(h_a)}) \langle \boldsymbol{h}|_{Sk} T_a^-, \tag{3.15}$$

*where[3]:*

$$\langle h_1, ..., h_a, ..., h_N|_{Sk} T_a^{\pm} = \langle h_1, ..., h_a \pm 1, ..., h_N|_{Sk}, \tag{3.16}$$

*while, the action of $C^{(K)}(\lambda)$ is uniquely defined by the quantum determinant relation.*

---

[2]We have introduced this requirement only to make easier the comparison with Theorem 4.2. In fact, the statement of the theorem holds also for non-invertible twist matrices, it is enough to remove the eigenvalue $k_1$ from the definition of the co-vectors (3.10) and (3.17).

[3]By convention the co-vectors $\langle h_1, \ldots, h_N|_{Sk}$ having an $h_i$ outside the set $\{0, \ldots, 2s_i\}$ are identically zero, which is of course compatible with the above actions of the operators $A^{(K)}(\lambda)$ and $D^{(K)}(\lambda)$, thanks to $a(\xi_a^{(2s_a)}) = 0$ and $d(\xi_a^{(0)}) = 0$.

*b) If* (3.3) *is verified and* $K \in \mathrm{End}(\mathbb{C}^2)$ *is any* $2 \times 2$ *matrix not proportional to the identity, invertible and such that* $b = 0$, *then the co-vectors* $\underline{\langle h|}_{Sk} \equiv \underline{\langle h_1, ..., h_N|}_{Sk}$, *defined by:*

$$\underline{\langle h|}_{Sk} \equiv \langle 0| \prod_{n=1}^{N} \prod_{k_n=0}^{h_n-1} \frac{A^{(\bar{K})}(\xi_n^{(k_n)})}{k_1 a(\xi_n^{(k_n)})} \mathcal{W}_K^{-1} \tag{3.17}$$

$$= \langle 0| \mathcal{W}_K^{-1} \prod_{n=1}^{N} \prod_{k_n=0}^{h_n-1} \frac{\tilde{A}^{(K)}(\xi_n^{(k_n)})}{k_1 a(\xi_n^{(k_n)})}, \tag{3.18}$$

*define a co-vector* $\tilde{B}^{(K)}$*-eigenbasis of* $\mathcal{H}$:

$$\underline{\langle h|}_{Sk} \tilde{B}^{(K)}(\lambda) = d_{\boldsymbol{h}}^{(\bar{K})}(\lambda) \underline{\langle h|}_{Sk}, \tag{3.19}$$

*where:*

$$d_{\boldsymbol{h}}^{(K)}(\lambda) \equiv \bar{b} \prod_{n=1}^{N} (\lambda - \xi_n^{(h_n)}) \quad and \quad \boldsymbol{h} \equiv (h_1, ..., h_N). \tag{3.20}$$

*Moreover it holds:*

$$\underline{\langle h|}_{Sk} \tilde{A}^{(K)}(\lambda) = \sum_{a=1}^{N} \prod_{b \neq a} \frac{\lambda - \xi_b^{(h_b)}}{\xi_a^{(h_a)} - \xi_b^{(h_b)}} k_1 a(\xi_a^{(h_a)}) \underline{\langle h|}_{Sk} T_a^+, \tag{3.21}$$

$$\underline{\langle h|}_{Sk} \tilde{D}^{(K)}(\lambda) = \sum_{a=1}^{N} \prod_{b \neq a} \frac{\lambda - \xi_b^{(h_b)}}{\xi_a^{(h_a)} - \xi_b^{(h_b)}} k_2 d(\xi_a^{(h_a)}) \underline{\langle h|}_{Sk} T_a^-. \tag{3.22}$$

*Proof.* The proof that the co-vectors (3.12) are eigenco-vectors of $B^{(K)}(\lambda)$ with the above defined eigenvalues is standard, see e.g. [44], it uses just the Yang-Baxter commutation relations and the fact that the left reference co-vector is a $B^{(K)}$-eigenco-vector. Indeed, it holds

$$A^{(K)}(\lambda) = aA(\lambda) + bC(\lambda), \quad B^{(K)}(\lambda) = aB(\lambda) + bD(\lambda), \tag{3.23}$$

$$C^{(K)}(\lambda) = cA(\lambda) + dC(\lambda), \quad D^{(K)}(\lambda) = cB(\lambda) + dD(\lambda), \tag{3.24}$$

and

$$\langle 0|A(\lambda) = a(\lambda)\langle 0|, \quad \langle 0|D(\lambda) = d(\lambda)\langle 0|, \quad \langle 0|B(\lambda) = \underline{0}, \quad \langle 0|C(\lambda) \neq \underline{0}. \tag{3.25}$$

Then the proof that the operators $A^{(K)}(\lambda)$ and $D^{(K)}(\lambda)$ have the given representation in the $B^{(K)}$-eigenco-vectors is once again a direct consequence of the Yang-Baxter commutation relations. Finally, note that the above construction generates

$$d_{\{s_n\}} = \prod_{n=1}^{N} (2s_n + 1), \tag{3.26}$$

i.e. the dimension of the representation, $B^{(K)}$-eigenco-vectors which are independent and so form a basis as soon as they are all nonzero as they are associated to different eigenvalues of $B^{(K)}(\lambda)$. This last statement can be for example shown by constructing the $B^{(K)}$-eigenvectors and proving that the action of a $B^{(K)}$-eigenco-vector on the $B^{(K)}$-eigenvector associated to the same eigenvalue is nonzero. We omit this steps as they can be done following exactly the same lines described in the case of the anti-periodic boundary conditions [44].

If the condition $b = 0$ is satisfied, then the above results allow similarly to show that the one parameter operator family $B^{(\bar{K})}(\lambda)$ is diagonalizable with simple spectrum and they allow

to derive its left eigenbasis. Then the same statements hold for the one parameter operator family $\tilde{\mathsf{B}}^{(K)}(\lambda)$ defining the SoV basis for $b = 0$, it being similar to $\mathsf{B}^{(\bar{K})}(\lambda)$:

$$
\underline{\langle \mathbf{h}|}_{Sk} \equiv \langle 0| \prod_{n=1}^{N} \prod_{k_n=0}^{h_n-1} \frac{\mathsf{A}^{(\bar{K})}(\xi_n^{(k_n)})}{\mathsf{k}_1 a(\xi_n^{(k_n)})} \mathcal{W}_K^{-1}, \tag{3.27}
$$

where:

$$
\underline{\langle \mathbf{h}|}_{Sk} \tilde{\mathsf{B}}^{(K)}(\lambda) = d_{\mathbf{h}}^{(K)}(\lambda) \underline{\langle \mathbf{h}|}_{Sk}. \tag{3.28}
$$

$\square$

# 4 New SoV bases and complete spectrum characterization

In this section we construct two different SoV bases from two natural sets of conserved charges of the considered models using the method developed in [1]. Moreover, we show how these SoV bases indeed separate the quantum spectral problem for the transfer matrix. We have already presented such a procedure [1, 2] in the case of models associated to fundamental representations of the rational Yang-Baxter algebra. Here we explain how this procedure can be developed for non-fundamental representations, using as an example higher spin representations of the rational $gl_2$ Yang-Baxter algebra.

In subsection 4.1, we introduce a first set of SoV co-vectors generated from the transfer matrices obtained from the fundamental Lax operators, i.e., the Lax operators for which the auxiliary space is isomorphic to its local quantum space at some site $n$. They are obtained by fusion from the original Lax operator having an auxiliary space in the spin-1/2 representation. In this case the proof that these conserved charges generate a basis of the Hilbert space is given following the same main steps used for the fundamental representations in [1]. This SoV basis is quite natural in this respect and it allows to prove also the simplicity of the transfer matrix spectrum and derive its complete characterization. In subsection 4.2, we then introduce another SoV basis constructed from the full tower of fused transfer matrices, which we argue to be the most natural with respect to the action of the transfer matrix that becomes explicitly linear in that basis thanks to the fusion rules satisfied by the quantum spectral invariants. There, we prove also that under some special choice of the generating co-vector this SoV basis coincides with Sklyanin's SoV basis presented in the previous section.

## 4.1 A first SoV basis construction and the associated spectrum characterization

The following proposition holds:

**Proposition 4.1.** *The following set of co-vectors, obtained from the action of the fundamental transfer matrices* $\mathsf{T}^{(K|2s_n)}(\xi_n^{(2s_n-1)})$

$$
{}_f\langle h_1, ..., h_N| \equiv \langle \Omega| \prod_{n=1}^{N} (\mathsf{T}^{(K|2s_n)}(\xi_n^{(2s_n-1)}))^{h_n} \quad \forall h_a \in \{0, ..., 2s_n\}, a \in \{1, ..., N\} \tag{4.1}
$$

*defines a co-vector basis of $\mathcal{H}$ for almost any choice of the co-vector $\langle \Omega|$ and of the inhomogeneity parameters $\{\xi_1, ..., \xi_N\}$ under the condition (3.3) and the requirement $K \in \text{End}(\mathbb{C}^2)$ diagonalizable, simple and invertible. In particular, we can chose the following tensor product form for the co-vector*

$$
\langle \Omega| \equiv \bigotimes_{n=1}^{N} \langle \omega_n|, \tag{4.2}
$$

*where* $\langle \omega_n |$ *is any local co-vector of* $V_n^{(2s_n)}$ *such that the set of* $2s_n + 1$ *co-vectors*

$$\langle \omega_n | \big[ K_n^{(2s_n)} \big]^{h_n} \text{ for } h_n \in \{0, ..., 2s_n\}, \tag{4.3}$$

*form a co-vector basis of* $V_n^{(2s_n)}$ *for any* $n \in \{1, ..., \mathsf{N}\}$. *Moreover, under the same conditions, the transfer matrix* $\mathsf{T}^{(K)}(\lambda)$ *is diagonalizable with simple spectrum.*

*Proof.* This is a corollary of the general Propositions 2.4 and 2.5 of our previous paper [1]. Indeed, the proof proceeds along the lines as in these propositions once we observe that the fused, fundamental, $R$-matrices satisfy the following identity [84]:

$$R_{ab}^{(2s|2s)}(-(s-1/2)\eta) = P_{ab} \in \text{End}(V_a^{(2s)} \otimes V_b^{(2s)}), \tag{4.4}$$

where $P_{ab} \in \text{End}(V_a^{(2s)} \otimes V_b^{(2s)})$ is the permutation operator for these spaces, hence leading to the following representation (the Lax operator being essentially proportional to the corresponding $R$-matrix in the same tensor product of representations):

$$\mathsf{T}^{(K|2s_n)}(\xi_n^{(2s_n-1)}) = \gamma R_{nn-1}^{(2s_n|2s_{n-1})}(\xi_n^{(2s_n-1)} - \xi_{n-1}) \cdots R_{n1}^{(2s_n|2s_1)}(\xi_n^{(2s_n-1)} - \xi_1) K_n^{(2s_n)}$$
$$\times R_{n\mathsf{N}}^{(2s_n|2s_\mathsf{N})}(\xi_n^{(2s_n-1)} - \xi_\mathsf{N}) \cdots R_{nn+1}^{(2s_n|2s_{n+1})}(\xi_n^{(2s_n-1)} - \xi_{n+1}), \tag{4.5}$$

for some non-zero constant $\gamma$. Moreover the leading asymptotic of any $R$-matrices $R_{nm}^{(2s_n|2s_m)}(\lambda)$ is proportional to the identity. Then, the existence of each $\langle \omega_n |$ is implied by the fact that the matrices $K_n^{(2s_n)}$ are all diagonalizable with simple spectrum. □

**Remark 1:** It is worth to point out that we have proven our proposition only in the case in which the twist matrix $K \in \text{End}(\mathbb{C}^2)$ has not only simple spectrum but is in addition diagonalizable and invertible. These requirements in the case of fundamental representations are not needed, here they are imposed to get that the fused twist matrices keep the simple spectrum nature which allows us to use the general Propositions 2.4 of our first paper [1]. It is then interesting to point out that in the next subsection 4.2 the SoV basis will be constructed in our approach using the commuting conserved charges without imposing these additional constraints, but just asking the simplicity of its spectrum which for a $2 \times 2$ matrix is equivalent to ask that it isn't proportional to the identity.

**Remark 2:** As explained in our previous papers once the SoV basis is constructed, by using the action of the quantum spectral invariants on some given generating co-vector, then the transfer matrix fusion equations allow for the full characterization of the transfer matrix spectrum. Indeed, the transfer matrix satisfies the property given in Proposition 2.3. It is interesting to point out that the Proposition 2.3 can be also derived as consequence of the SoV characterization of the transfer matrix spectrum. Indeed, in Sklyanin's framework, one can prove that any transfer matrix eigenvalue has to satisfy the discrete system of equations (4.8) written bellow by computing the action of the transfer matrix on the generic eigenvector in the SoV representation, as it was done in the anti-periodic case in [44]. Then, the Proposition 2.3 holds for any diagonalizable and simple spectrum twist matrix; indeed by Proposition 4.1 the transfer matrix share the same properties and then it satisfies the system of equations (2.37). Now it is enough to observe that the determinants on the l.h.s. of (2.37) are polynomials in the elements of the twist matrix to derive that (2.37) has to hold for any twist matrix as it holds for almost any value of its elements.

Here, instead, we use the Proposition 2.3 to prove that any solution to this system of equations indeed generates an eigenvalue and eigenvector in our SoV basis. Indeed, let us define the function:

$$g_a(\lambda) = \prod_{b \neq a, b=1}^{\mathsf{N}} \frac{\lambda - \xi_b^{(0)}}{\xi_a^{(0)} - \xi_b^{(0)}}, \tag{4.6}$$

then we can state the following:

**Theorem 4.1.** *Under the same conditions allowing to define the SoV basis* (4.1)*, the spectrum of* $\mathsf{T}^{(K)}(\lambda)$ *coincides with the set of polynomials:*

$$\Sigma_{\mathsf{T}^{(K)}} = \left\{ t(\lambda) : t(\lambda) = tr\, K \prod_{a=1}^{\mathsf{N}}(\lambda - \xi_n^{(0)}) + \sum_{a=1}^{\mathsf{N}} g_a(\lambda) x_a, \quad \forall\{x_1,...,x_{\mathsf{N}}\} \in D_{\mathsf{T}^{(K)}} \right\}, \quad (4.7)$$

*where* $D_{\mathsf{T}^{(K)}}$ *is the set of* $\mathsf{N}$*-tuples* $\{x_1,...,x_{\mathsf{N}}\}$ *solutions to the following system of* $\mathsf{N}$ *equations:*

$$\det_{2s_n+1} D_{t,n} = 0, \quad \forall n \in \{1,...,\mathsf{N}\}, \quad (4.8)$$

*each of which is a degree* $2s_n + 1$ *polynomial equation in the* $\mathsf{N}$ *unknowns* $\{x_1,...,x_{\mathsf{N}}\}$ *for any fixed n. Here, we have defined:*

$$D_{t,n} \equiv$$

$$\begin{pmatrix}
t(\xi_n^{(0)}) & -k_1 a(\xi_n^{(0)}) & 0\dots & & 0 & 0 \\
-k_2 d(\xi_n^{(1)}) & t(\xi_n^{(1)}) & -k_1 a(\xi_n^{(1)})\dots & & 0 & 0 \\
0 & \ddots & & & & \\
\vdots & & \ddots & & & \\
\vdots & & 0\dots & -k_2 d(\xi_n^{(2s_n-1)}) & t(\xi_n^{(2s_n-1)}) & -k_1 a(\xi_n^{(2s_n-1)}) \\
0 & \dots & 0\dots & 0 & -k_2 d(\xi_n^{(2s_n)}) & t(\xi_n^{(2s_n)})
\end{pmatrix}.$$

$$(4.9)$$

*Furthermore, for any* $t(\lambda) \in \Sigma_{\mathsf{T}^{(K)}}$ *the following factorized wavefunction in the SoV co-vector basis*[4]:

$$_f\langle h_1,...,h_N|t\rangle = \prod_{n=1}^{\mathsf{N}} \left( \det_{2s_n} D_{t,n}^{(2s_n+1,2s_n+1)} \right)^{h_n}, \quad (4.10)$$

*characterizes the associated unique eigenvector* $|t\rangle$*, up-to an overall normalization that we have fixed by* $\langle\Omega|t\rangle = 1$.

*Proof.* From the Proposition 2.3 it follows that any eigenvalue $t(\lambda) \in \Sigma_{\mathsf{T}^{(K)}}$ is indeed a degree $\mathsf{N}$ polynomial in $\lambda$ with central asymptotics (2.36) and solutions of the system (2.37) as a consequence of the fusion equations.

We have now to prove the reverse statement that any solution of this system of equations $\{x_1,...,x_{\mathsf{N}}\} \in D_{\mathsf{T}^{(K)}}$ indeed define through the polynomial interpolation formula (4.7) a transfer matrix eigenvalue. The very existence of the SoV basis, for almost any value of the inhomogeneities, has as a direct consequence that the spectrum of all transfer matrices $\mathsf{T}^{(K|2s_n)}$ is simple. All these fused transfer matrices being polynomials in the transfer matrix it implies (otherwise we get an immediate contradiction) that $\mathsf{T}^{(K)}(\lambda)$ has simple spectrum and it is diagonalizable. Indeed, the Proposition 2.5 of our first paper [1] applies here too. So that there exists $d_{\{s_n\}}$ different eigenvalues $t(\lambda) \in \Sigma_{\mathsf{T}^{(K)}}$, i.e. a number equal to the dimension of the representation.

It is easy now to remark that, as the theorem states, (4.8) is a system of $\mathsf{N}$ polynomial equations in the $\mathsf{N}$ unknowns $\{x_{a\leq\mathsf{N}}\}$ of degree $2s_n+1$ for any $n \in \{1,...,\mathsf{N}\}$. Then, by the Theorem of Bèżout [96], we can distinguish two cases: i. the $\mathsf{N}$ polynomials of degree $2s_n + 1$ in the $\mathsf{N}$

---

[4]Here, we are using the notation $D_{t,n}^{(i,j)}$ to represent the $2s_n \times 2s_n$ matrix obtained by removing the line $i$ and the column $j$ from the matrix $D_{t,n}$.

variables $\{x_1, ..., x_N\}$, defining the system, have common components, i.e. one or more common roots, then the system admits infinite number of solutions $\{x_1, ..., x_N\}$. ii. these N polynomials do not contain common components, then the system admits a finite number of solutions $\{x_1, ..., x_N\}$ which counted with their multiplicity coincides with $d_{\{s_n\}}$. As we have already shown that to any $t(\lambda) \in \Sigma_{T^{(K)}}$ it is uniquely associated a solution $\{x_{a \leq N} \equiv t(\xi_{a \leq N}^{(0)})\} \in D_{T^{(K)}}$, then we can state that the system (4.8) admits at least $d_{\{s_n\}}$ distinct solutions. Then, once the condition of no common components is satisfied, we derive that the system admits exactly $d_{\{s_n\}}$ distinct solutions and each one is associated to a transfer matrix eigenvalue, which complete the equivalence proof.

So we are left with the proof of this condition of no common components. In order to prove this statement it is enough to show it for some special value of the twist eigenvalues to imply its validity for almost any value of these parameters, as the polynomials defining the system are polynomial in the twist parameters too. Let us here consider the case $k_1 \neq 0$ and $k_2 = 0$, then the system of equations reads:

$$\prod_{h=0}^{2s_n} t(\xi_n^{(h)}) = 0 \quad \forall n \in \{1, ..., N\}. \tag{4.11}$$

Now taking into account that by definition $t(\lambda)$ is a degree N polynomial in $\lambda$ and that it holds:

$$\xi_n^{(h)} \neq \xi_m^{(k)} \quad \forall h \in \{0, ..., 2s_n\}, k \in \{0, ..., 2s_m\}, n \neq m \in \{1, ..., N\}, \tag{4.12}$$

by the condition (3.3), then we have that a solution to the system can be realized iff for any $n \in \{1, ..., N\}$ there exists a unique $h_n \in \{0, ..., 2s_n\}$ such that $t(\xi_n^{(h_n)}) = 0$, or equivalently:

$$t_{\mathbf{h}}(\lambda) = k_1 \prod_{n=1}^{N} (\lambda - \xi_n^{(h_n)}). \tag{4.13}$$

So we have that the system has exactly $d_{\{s_n\}}$ distinct solutions, defined by fixing the $d_{\{s_n\}}$ distinct N-upla $\mathbf{h} = \{h_1, ..., h_N\}$ with $h_n \in \{0, ..., 2s_n\}$, $n = 1, ..., N$, in this way implying the condition of no common components.

Finally, by definition of the SoV basis for any $t(\lambda) \in \Sigma_{T^{(K)}}$ the uniquely associated eigenvector has the factorized wavefunctions:

$$_f\langle h_1, ..., h_N | t \rangle = \prod_{n=1}^{N} \left( t^{(2s_n)}(\xi_n^{(2s_n-1)}) \right)^{h_n}, \tag{4.14}$$

where the polynomial $t^{(2s_n)}(\lambda)$ is the eigenvalue of the fused transfer matrix $T^{(K|2s_n)}(\lambda)$ uniquely defined in terms of the $t(\lambda) \in \Sigma_{T^{(K)}}$ by the use of the fusion equations. Then the statement of the theorem follows observing that the following identities:

$$t^{(2s_n)}(\xi_n^{(2s_n-1)}) = \det_{2s_n} D_{t,n}^{(2s_n+1,2s_n+1)}, \tag{4.15}$$

are direct consequences of the resolution of the fusion relations (2.25), the definition (4.1) and the fact that

$$D_{2s_n+1}(T^{(K)}(\xi_n^{(2s_n)}))|t\rangle = D_{t,n}|t\rangle \tag{4.16}$$

together with $D_{l+1}(T^{(K)}(\lambda))^{(l+1,l+1)} = D_l(T^{(K)}(\lambda + \eta))$ and $\xi_n^{(2s_n-1)} = \xi_n^{(2s_n)} + \eta$.

$\square$

## 4.2 The natural SoV basis and the associated spectrum characterization

Here we show that there exists a different choice of the SoV basis for which the action of the transfer matrix becomes very simple as a consequence of the fusion relations and in particular of the Proposition 2.3.

**Theorem 4.2.** *The set consisting of the co-vectors*

$$\langle h_1,...,h_N | \equiv \langle S | \prod_{n=1}^{N} \frac{k_2^{h_n-2s_n} T^{(K|2s_n-h_n)}(\xi_n^{(2s_n)})}{\prod_{k=0}^{2s_n-h_n-1} d(\xi_n^{(2s_n-k)})} \quad \forall h_a \in \{0,...,2s_a\}, a \in \{1,...,N\} \qquad (4.17)$$

*defines a co-vector basis of $\mathcal{H}$ for almost any choice of the co-vector $\langle S |$ and of the inhomogeneity parameters $\{\xi_1,...,\xi_N\}$ satisfying (3.3) under the condition $K \in \mathrm{End}(\mathbb{C}^2)$ not proportional to the identity and invertible[5]. Moreover, denoting $\langle O | = \langle 0...,0 |$ the co-vector $\langle h_1,...,h_N |$ of this basis when all $h_i = 0$, we also have, using the higher fusion relation (2.38):*

$$\langle h_1,...,h_N | \equiv \langle O | \prod_{n=1}^{N} \frac{T^{(K|h_n)}(\xi_n^{(h_n-1)})}{k_1^{h_n} \prod_{k=0}^{h_n-1} a(\xi_n^{(k)})} \quad \forall h_a \in \{0,...,2s_a\}, a \in \{1,...,N\}. \qquad (4.18)$$

*Furthermore, we have:*
   *a) if $b \neq 0$, fixing*

$$\langle S | \equiv \langle h_1 = 2s_1,...,h_N = 2s_N |_{Sk}, \qquad (4.19)$$

*meaning also that $\langle O | = \langle 0 |$, then our SoV basis coincides with Sklyanin's type SoV basis, i.e. it holds:*

$$\langle h_1,...,h_N |_{Sk} = \langle h_1,...,h_N | \quad \forall h_n \in \{0,...,2s_n\}, n \in \{1,...,N\}. \qquad (4.20)$$

   *b) if $b = 0$, fixing*

$$\langle S | \equiv \underline{\langle h_1 = 2s_1,...,h_N = 2s_N |}_{Sk}, \qquad (4.21)$$

*then our SoV basis coincides with Sklyanin's type SoV basis, i.e. it holds:*

$$\underline{\langle h_1,...,h_N |}_{Sk} = \langle h_1,...,h_N | \quad \forall h_n \in \{0,...,2s_n\}, n \in \{1,...,N\}. \qquad (4.22)$$

*Proof.* Let us remark that the determinant of the $d_{\{s_n\}} \times d_{\{s_n\}}$ matrix whose columns are the elements of the co-vectors (4.17) in the natural basis is a polynomial of order at most $d_{\{s_n\}}$ in the components of the co-vector $\langle S |$. So that it is enough to prove that this determinant is nonzero for a special choice of $\langle S |$ to show that it is nonzero for almost any choice of $\langle S |$. So to prove the theorem it is enough to prove that the statements a) and b) hold.

Let us first observe that, by using the quantum determinant condition, we have the following rewriting of Sklyanin's type SoV basis:

$$\langle h_1,...,h_N |_{Sk} = \langle S | \prod_{n=1}^{N} \prod_{k_n=h_n+1}^{2s_n} \frac{D^{(K)}(\xi_n^{(k_n)})}{d(\xi_n^{(k_n)}) k_2}, \qquad (4.23)$$

and

$$\underline{\langle h_1,...,h_N |}_{Sk} = \langle S | \prod_{n=1}^{N} \prod_{k_n=h_n+1}^{2s_n} \frac{D^{(\bar{K})}(\xi_n^{(k_n)})}{d(\xi_n^{(k_n)}) k_2} \mathcal{W}_K^{-1}. \qquad (4.24)$$

So let us prove now the statement a). The proof is done by induction, in fact it holds if $h_i = 2s_i$ for all the $i \in \{1,...,N\}$, by the given choice of the co-vector $\langle S |$. So we assume that our

---

[5]It is worth mentioning that the construction of the SoV basis both in Sklyanin's framework and in our current one can be developed also for non-invertible simple twist matrices. However, we use here this additional requirement just to present a simpler form of the theorem and its proof.

statement holds for any N-upla $\{h_{i\leq N}\}$ such that $h_i \geq \bar{h}_i \geq 1$ for some fixed N-upla $\{\bar{h}_1, ..., \bar{h}_N\}$ and then we prove that it holds for the N-upla $\{h_1, ..., \bar{h}_n - 1, ..., h_N\}$ for any $n \in \{1, ..., N\}$. In order to do so, we expand the following co-vector:

$$\langle h_1, ..., \bar{h}_n - 1, ..., h_N| \equiv \langle h_1, ..., \hat{h}_n = 2s_n, ..., h_N| \frac{\mathsf{k}_2^{(\bar{h}_n-(1+2s_n))} \mathsf{T}^{(K|2s_n+1-\bar{h}_n)}(\xi_n^{(2s_n)})}{\prod_{k=0}^{2s_n-\bar{h}_n} d(\xi_n^{(2s_n-k)})}$$

$$= \langle h_1, ..., \hat{h}_n = 2s_n, ..., h_N| \frac{\mathsf{k}_2^{(\bar{h}_n-(1+2s_n))}}{\prod_{k=0}^{2s_n-\bar{h}_n} d(\xi_n^{(2s_n-k)})}$$

$$\times (\mathsf{T}^{(K)}(\xi_n^{(\bar{h}_n)}) \mathsf{T}^{(K|2s_n-\bar{h}_n)}(\xi_n^{(2s_n)}) - \Delta_\eta^{(K)}(\xi_n^{(\bar{h}_n)}) \mathsf{T}^{(K|2s_n-(\bar{h}_n+1))}(\xi_n^{(2s_n)})), \quad (4.25)$$

where for any $i \in \{1, ..., N\}\setminus\{n\}$ we have fixed $h_i \geq \bar{h}_i \geq 1$, which by definition implies:

$$\langle h_1, ..., \bar{h}_n - 1, ..., h_N| = \langle h_1, ..., \bar{h}_n, ..., h_N| \frac{\mathsf{T}^{(K)}(\xi_n^{(\bar{h}_n)})}{d(\xi_n^{(\bar{h}_n)})\mathsf{k}_2}$$

$$- \frac{\Delta_\eta^{(K)}(\xi_n^{(\bar{h}_n)})}{\mathsf{k}_2^2 d(\xi_n^{(\bar{h}_n+1)}) d(\xi_n^{(\bar{h}_n)})} \langle h_1, ..., \bar{h}_n + 1, ..., h_N|. \quad (4.26)$$

Then the induction hypothesis implies that it holds:

$$\langle h_1, ..., \bar{h}_n - 1, ..., h_N| = \langle h_1, ..., \bar{h}_n, ..., h_N|_{Sk} \frac{\mathsf{T}^{(K)}(\xi_n^{(\bar{h}_n)})}{d(\xi_n^{(\bar{h}_n)})\mathsf{k}_2}$$

$$- \frac{\Delta_\eta^{(K)}(\xi_n^{(\bar{h}_n)})}{\mathsf{k}_2^2 d(\xi_n^{(\bar{h}_n+1)}) d(\xi_n^{(\bar{h}_n)})} \langle h_1, ..., \bar{h}_n + 1, ..., h_N|_{Sk}. \quad (4.27)$$

Now by definition of the transfer matrix and Sklyanin's type SoV basis we have that it holds:

$$\langle h_1, ..., \bar{h}_n, ..., h_N|_{Sk} \frac{\mathsf{T}^{(K)}(\xi_n^{(\bar{h}_n)})}{d(\xi_n^{(\bar{h}_n)})\mathsf{k}_2} = \langle h_1, ..., \bar{h}_n - 1, ..., h_N|_{Sk}$$

$$+ \langle h_1, ..., \bar{h}_n + 1, ..., h_N|_{Sk} \frac{\mathsf{D}^{(K)}(\xi_n^{(\bar{h}_n+1)}) \mathsf{A}^{(K)}(\xi_n^{(\bar{h}_n)})}{\mathsf{k}_2^2 d(\xi_n^{(\bar{h}_n+1)}) d(\xi_n^{(\bar{h}_n)})}. \quad (4.28)$$

So that by using the quantum determinant identity:

$$\Delta_\eta^{(K)}(\lambda) = \mathsf{D}^{(K)}(\lambda - \eta) \mathsf{A}^{(K)}(\lambda) - \mathsf{B}^{(K)}(\lambda - \eta) \mathsf{C}^{(K)}(\lambda) \quad (4.29)$$

it holds:

$$\langle h_1, ..., \bar{h}_n + 1, ..., h_N|_{Sk} \frac{\mathsf{D}^{(K)}(\xi_n^{(\bar{h}_n+1)}) \mathsf{A}^{(K)}(\xi_n^{(\bar{h}_n)})}{\mathsf{k}_2^2 d(\xi_n^{(\bar{h}_n+1)}) d(\xi_n^{(\bar{h}_n)})}$$

$$= \langle h_1, ..., \bar{h}_n + 1, ..., h_N|_{Sk} \frac{\Delta_\eta^{(K)}(\xi_n^{(\bar{h}_n)})}{\mathsf{k}_2^2 d(\xi_n^{(\bar{h}_n+1)}) d(\xi_n^{(\bar{h}_n)})}, \quad (4.30)$$

being

$$\langle h_1, ..., \bar{h}_n + 1, ..., h_N|_{Sk} \frac{\mathsf{B}^{(K)}(\xi_n^{(\bar{h}_n+1)}) \mathsf{C}^{(K)}(\xi_n^{(\bar{h}_n)})}{\mathsf{k}_2^2 d(\xi_n^{(\bar{h}_n+1)}) d(\xi_n^{(\bar{h}_n)})} = 0, \quad (4.31)$$

by definition of Sklyanin's type SoV basis. So that replacing these results in (4.27) we get our identity:

$$\langle h_1,...,\bar{h}_n-1,...,h_N| \equiv \langle h_1,...,\bar{h}_n-1,...,h_N|_{Sk}, \tag{4.32}$$

for any $h_i \geq \bar{h}_i \geq 1$ with $i \in \{1,...,N\}\backslash\{n\}$, which proves the induction. Note that the second representation, using $\langle 0|$ as starting co-vector, can be either proven directly using the higher fusion relation (2.38) or again along a similar proof as above using recursively the standard fusion relations. In the case b), following the same proof of case a) for the transfer matrices

$$T^{(\bar{K}|l)}(\lambda) = \mathcal{W}_K^{-1} T^{(K|l)}(\lambda) \mathcal{W}_K \tag{4.33}$$

we show the identities:

$$\langle h_1,...,h_N|_{Sk} \mathcal{W}_K = \langle h_1 = 2s_1,...,h_N = 2s_N|_{Sk} \mathcal{W}_K \prod_{n=1}^{N} \frac{k_2^{(h_n-2s_n)} T^{(\bar{K}|2s_n-h_n)}(\xi_n^{(2s_n)})}{\prod_{k=0}^{2s_n-(h_n+1)} d(\xi_n^{(2s_n-k)})} \tag{4.34}$$

for any $h_n \in \{0,...,2s_n\}$ and $n \in \{1,...,N\}$ from which our statement easily follows. $\square$

As we have anticipated before this is the natural SoV basis as the action of the transfer matrix on it is easily derived by the fusion identities. Indeed, we have the following:

**Proposition 4.2.** *The transfer matrix $T^{(K)}(\lambda)$ has the following separate action on the SoV basis co-vectors:*

$$\langle h_1,...,h_N|T^{(K)}(\xi_n^{(h_n)}) = k_1 a(\xi_n^{(h_n)}) \langle h_1,...,h_n+1,...,h_N| + k_2 d(\xi_n^{(h_n)}) \langle h_1,...,h_n-1,...,h_N|. \tag{4.35}$$

*Proof.* Let us compute:

$$\langle h_1,...,h_N|T^{(K)}(\xi_n^{(h_n)}), \tag{4.36}$$

by definition it can be rewritten as:

$$\langle h_1,...,\hat{h}_n=2s_n,...,h_N| \frac{k_2^{(h_n-2s_n)} T^{(K|2s_n-h_n)}(\xi_n^{(2s_n)})}{\prod_{k=0}^{2s_n-(h_n+1)} d(\xi_n^{(2s_n-k)})} T^{(K)}(\xi_n^{(h_n)}), \tag{4.37}$$

now we can use the fusion relations to rewrite the following operator product:

$$T^{(K)}(\xi_n^{(h_n)}) T^{(K|2s_n-h_n)}(\xi_n^{(2s_n)}) = \Delta_\eta^{(K)}(\xi_n^{(h_n)}) T^{(K|2s_n-(h_n+1))}(\xi_n^{(2s_n)}) + T^{(K|2s_n-(h_n-1))}(\xi_n^{(2s_n)}), \tag{4.38}$$

so that using the l.h.s. we get:

$$\langle h_1,...,h_N|T^{(K)}(\xi_n^{(h_n)})$$
$$= \langle h_1,...,\hat{h}_n=2s_n,...,h_N|k_1 a(\xi_n^{(h_n)}) \frac{k_2^{(h_n+1-2s_n)} T^{(K|2s_n-(h_n+1))}(\xi_n^{(2s_n)})}{\prod_{k=0}^{2s_n-(h_n+2)} d(\xi_n^{(2s_n-k)})}$$
$$+ \langle h_1,...,\hat{h}_n=2s_n,...,h_N|k_2 d(\xi_n^{(h_n)}) \frac{k_2^{(h_n-1-2s_n)} T^{(K|2s_n-(h_n-1))}(\xi_n^{(2s_n)})}{\prod_{k=0}^{2s_n-h_n} d(\xi_n^{(2s_n-k)})} \tag{4.39}$$

from which our statement follows by the definition of the SoV basis. Again here, we could also start from the second representation having $\langle O|$ as starting co-vector, and get from there the same result. $\square$

From the above proposition, we get the following characterization of the transfer matrix spectrum in our new SoV basis. Clearly the discrete characterization of the transfer matrix eigenvalues is independent of the chosen SoV basis so it coincides with the one we have given in the theorem of the previous section. Of course what changes is the characterization of the transfer matrix eigenvectors in the new SoV basis; indeed, it holds:

**Theorem 4.3.** *Let us assume $K \in \text{End}(\mathbb{C}^2)$ not proportional to the identity and invertible, then for almost any value of the inhomogeneities $\mathsf{T}^{(K)}(\lambda)$ has simple spectrum, is diagonalizable and the set of its eigenvalues $\Sigma_{\mathsf{T}^{(K)}}$ coincides with the set of degree $\mathsf{N}$ polynomials (4.7). For any $t(\lambda) \in \Sigma_{\mathsf{T}^{(K)}}$ the associated unique (up-to normalization fixed by $\langle S|t\rangle = 1$) eigenvector $|t\rangle$ has the following factorized wavefunction in the SoV co-vector basis:*

$$\langle h_1, ..., h_N|t\rangle = \prod_{n=1}^{N} \frac{\mathsf{k}_2^{(h_n - 2s_n)} t^{(2s_n - h_n)}(\xi_n^{(2s_n)})}{\prod_{k=0}^{2s_n - (h_n + 1)} d(\xi_n^{(2s_n - k)})}, \tag{4.40}$$

*where $t^{(2s_n - h_n)}(\xi_n^{(2s_n)})$ is the eigenvalue of the fused transfer matrix $T^{(2s_n - h_n)}(\xi_n^{(2s_n)})$ associated to the eigenvector $|t\rangle$.*

*Proof.* The proof can be done along the same lines as in the theorem of the previous section. It is however interesting to point out that in the current SoV basis the direct action of the transfer matrix allows to provide a simple and alternative proof of the fact that any solution $\{x_1, ..., x_N\} \in D_{\mathsf{T}^{(K)}}$ defines a transfer matrix eigenvalue through the polynomial interpolation formula (4.7). Indeed, the following identity:

$$\langle h_1, ..., h_N|\mathsf{T}^{(K)}(\xi_n^{(h_n)})|t\rangle = t(\xi_n^{(h_n)})\langle h_1, ..., h_N|t\rangle, \ \forall h_n \in \{0, ..., 2s_n\}, n \in \{1, ..., N\}, \tag{4.41}$$

is trivially deduced from the definition (4.40) of the state $|t\rangle$ and the previous proposition on the action of the transfer matrix on the SoV basis. Then the above results together with the asymptotics of the polynomials (4.7) imply our statement. $\square$

# 5 Q-operator and quantum spectral curve

## 5.1 The quantum spectral curve

A functional equation which provides an equivalent characterization of the SoV discrete characterization of the spectrum we derived above, the so-called quantum spectral curve, is given here. In the case at hand it is a second order Baxter difference equation.

**Theorem 5.1.** *Let the twist matrix $K \in End(\mathbb{C}^2)$ be such that[6] $\mathsf{k}_1 \neq \mathsf{k}_2$, $\mathsf{k}_1 \neq 0$, $\mathsf{k}_2 \neq 0$ and the inhomogeneities $\{\xi_1, ..., \xi_N\} \in \mathbb{C}^N$ satisfy the condition (3.3). Then an entire function $t(\lambda)$ is an element of $\Sigma_{\mathsf{T}^{(K)}}$ iff there exists a unique polynomial:*

$$Q_t(\lambda) = \prod_{a=1}^{M}(\lambda - \lambda_a), \text{ with } \mathsf{M} \le \mathsf{N}_s \equiv 2\sum_{n=1}^{N} s_n, \tag{5.1}$$

*such that $\lambda_a \neq \xi_b^{(2s_b)}$ for any $(a, b) \in \{1, ..., M\} \times \{1, ..., N\}$, satisfying the following quantum spectral curve functional equation:*

$$\mathsf{k}_1^2 a(\lambda)a(\lambda - \eta)Q_t(\lambda - 2\eta) - \mathsf{k}_1 a(\lambda)t(\lambda - \eta)Q_t(\lambda - \eta) + \Delta_\eta^{(K)}(\lambda)Q_t(\lambda) = 0. \tag{5.2}$$

---

[6]Note that the reformulation in terms of functional equations indeed hold also for the cases $(\mathsf{k}_2 = 0, \mathsf{k}_1 \neq 0)$ and $(\mathsf{k}_1 = 0, \mathsf{k}_2 \neq 0)$. Note that for these cases both the spectrum of the transfer matrix and of the associated $Q$-functions are explicitly known, see the proof of this theorem.

*Equivalently, shifting $\lambda$ by $\eta$ and dividing by common factor polynomials, it can be written as:*

$$t(\lambda)Q_t(\lambda) = \mathsf{k}_1 a(\lambda)Q_t(\lambda-\eta) + \mathsf{k}_2 d(\lambda)Q_t(\lambda+\eta). \tag{5.3}$$

*Up to an overall normalization the associated transfer matrix eigenvector $|t\rangle$ admits the following rewriting in the left SoV basis:*

$$\langle h_1,...,h_\mathsf{N}|t\rangle = \prod_{n=1}^{\mathsf{N}} Q_t(\xi_n^{(h_n)}). \tag{5.4}$$

*Proof.* Let us start proving that the quantum spectral curve equation admits at most one polynomial solution $Q_t(\lambda)$ for a given function $t(\lambda)$. Indeed, if we assume the existence of two such polynomial solutions $P(\lambda)$ and $Q(\lambda)$, it holds:

$$\frac{\mathsf{k}_1 a(\lambda)P(\lambda-\eta) + \mathsf{k}_2 d(\lambda)P(\lambda+\eta)}{P(\lambda)} = \frac{\mathsf{k}_1 a(\lambda)Q(\lambda-\eta) + \mathsf{k}_2 d(\lambda)Q(\lambda+\eta)}{Q(\lambda)}, \tag{5.5}$$

which can be rewritten

$$\mathsf{k}_1 a(\lambda) W_{P,Q}(\lambda) = \mathsf{k}_2 d(\lambda) W_{P,Q}(\lambda+\eta). \tag{5.6}$$

$W_{P,Q}(\lambda)$ is the quantum Wronskian of these two solutions:

$$W_{P,Q}(\lambda) = Q(\lambda)P(\lambda-\eta) - P(\lambda)Q(\lambda-\eta). \tag{5.7}$$

Taking into account that the zeroes of $d(\lambda)$ coincides with those of $a(\lambda)$ shifted by $2s_n\eta$ for any $n \in \{1,...,\mathsf{N}\}$ it follows:

$$W_{P,Q}(\lambda) = w_{P,Q}(\lambda) \prod_{n=1}^{\mathsf{N}} \prod_{h_n=0}^{2s_n-1} (\lambda - \xi_n^{(h_n)}), \tag{5.8}$$

where $w_{P,Q}(\lambda)$ is a polynomial in $\lambda$ which moreover has to satisfy the following quasi-periodicity condition:

$$\mathsf{k}_1 w_{P,Q}(\lambda+\eta) = \mathsf{k}_2 w_{P,Q}(\lambda). \tag{5.9}$$

Being $\mathsf{k}_1 \neq \mathsf{k}_2$ this implies[7] $w_{P,Q}(\lambda) = 0$.

Let us now assume the existence of $Q_t(\lambda)$ satisfying with $t(\lambda)$ the functional equation (5.2), then it follows that $t(\lambda)$ is a polynomial of degree $\mathsf{N}$ with leading coefficient $t_{\mathsf{N}+1}$ satisfying the equation:

$$\mathsf{k}_1^2 - \mathsf{k}_1 t_{\mathsf{N}+1} + \det K = 0, \tag{5.10}$$

which imposes $t_{\mathsf{N}+1} = \mathrm{tr}K = \mathsf{k}_1 + \mathsf{k}_2$. It is now easy to verify that particularizing the functional equation in the points $\lambda = \xi_n^{(h_n)} + \eta$, for any $h_n \in \{0,...,2s_n\}$, then the condition $Q_t(\xi_n^{(2s_n)}) \neq 0$ implies that $t(\lambda)$ satisfies the equation (4.8) for any fixed $n \in \{1,...,\mathsf{N}\}$ which together with its asymptotic behavior implies that $t(\lambda)$ is a transfer matrix eigenvalue.

The reverse statement is proven now. Let us assume that $t(\lambda)$ is a transfer matrix eigenvalue then we can show the existence of the polynomial $Q_t(\lambda)$ of the form (5.1) satisfying the required functional equation. On the l.h.s. of the equation there is a polynomial in $\lambda$ of maximal degree $2\mathsf{N} + M$, with $M \leq \mathsf{N}_s$, so that in order to prove that the functional equation is satisfied we have to show that it is zero in $2\mathsf{N} + \mathsf{N}_s + 1$ different points. First, at infinity, thanks to (5.10), the leading coefficient of this polynomial is zero as $t(\lambda)$ is a transfer matrix

---

[7]It is enough to compare the leading coefficients of the polynomial expansion to deduce that the equation is never satisfied for any nonzero polynomial.

eigenvalue. It is easy to remark that in the N points $\xi_a^{(2s_a)}$, for any $a \in \{1, ..., N\}$, the functional equation is automatically satisfied. Finally, it is satisfied in the $N + N_s$ points $\xi_a^{(h_a)}$ for any $h_a \in \{-1, ..., 2s_a - 1\}$ and $a \in \{1, ..., N\}$ if the homogeneous system:

$$
D_{t,n} \begin{pmatrix} Q_t(\xi_a^{(0)}) \\ Q_t(\xi_a^{(1)}) \\ \vdots \\ \vdots \\ Q_t(\xi_a^{(2s_n)}) \end{pmatrix}_{(2s_n+1)} = \begin{pmatrix} 0 \\ \vdots \\ \vdots \\ \vdots \\ 0 \end{pmatrix}_{(2s_n+1)} \quad \forall n \in \{1, ..., N\} \tag{5.11}
$$

is satisfied. As a consequence of the fact that $t(\lambda)$ is an eigenvalue we have:

$$
\det_{2s_n+1} D_{t,n} = 0 \quad \forall n \in \{1, ..., N\}, \tag{5.12}
$$

so that the previous system is equivalent (for example) to the system of the last $2s_n$ equations which can be resolved in terms of $Q_t(\xi_n^{(2s_n)})$ as it follows:

$$
Q_t(\xi_n^{(h_n)}) = Q_{t,n}^{(h_n)} Q_t(\xi_n^{(2s_n)}), \quad \forall h_n \in \{0, ..., 2s_n - 1\}, \tag{5.13}
$$

where we define $Q_{t,n}^{(2s_n)} = 1$ and the others, due to the tridiagonal form of the matrix $D_{t,n}$, are defined in a unique way recursively by:

$$
Q_{t,n}^{(h_n-1)} = \frac{t(\xi_n^{(h_n)}) Q_{t,n}^{(h_n)}}{k_2 d(\xi_n^{(h_n)})} - \frac{k_1 a(\xi_n^{(h_n)}) Q_{t,n}^{(h_n+1)}}{k_2 d(\xi_n^{(h_n)})}, \quad \forall h_n \in \{1, ..., 2s_n - 1\}, \tag{5.14}
$$

$$
Q_{t,n}^{(2s_n-1)} = t(\xi_n^{(2s_n)})/k_2 d(\xi_n^{(2s_n)}). \tag{5.15}
$$

Due to the tridiagonal form of the matrix $D_{t,n}$ these recursion relations can be solved explicitly in terms of the transfer matrix eigenvalues in a very simple way as:

$$
Q_{t,n}^{(2s_n-h_n)} = \frac{k_2^{-h_n} t^{(h_n)}(\xi_n^{(2s_n)})}{\prod_{k=0}^{h_n-1} d(\xi_n^{(2s_n-k)})}, \tag{5.16}
$$

hence leading to the result (5.4) by direct comparison with (4.40) with the convention that $t^{(1)}(\lambda) = t(\lambda)$. The proof can be done by an elementary induction. Indeed, the formula holds for $h_n = 1$ from the relation (5.15). Suppose the formula is true up to a value $l \in \{1, ..., 2s_n - 2\}$ then let us prove it for $l + 1$. By using (5.14) and then applying the induction hypothesis for $l$ and $l - 1$, we have:

$$
\begin{aligned}
Q_{t,n}^{(2s_n-(l+1))} &= \frac{t(\xi_n^{(2s_n-l)})(Q_{t,n}^{(2s_n-l)}}{k_2 d(\xi_n^{(2s_n-l)})} - \frac{k_1 a(\xi_n^{(2s_n-l)}) Q_{t,n}^{(2s_n-(l-1))}}{k_2 d(\xi_n^{(2s_n-l)})} \\
&= \frac{k_2^{-l} t(\xi_n^{(2s_n-l)}) t^{(l)}(\xi_n^{(2s_n)}) - k_1 a(\xi_n^{(2s_n-l)}) d(\xi_n^{(2s_n-l+1)}) k_2^{-(l-1)} t^{(l-1)}(\xi_n^{(2s_n)})}{k_2 d(\xi_n^{(2s_n-l)}) \prod_{k=0}^{l-1} d(\xi_n^{(2s_n-k)})} \\
&= \frac{k_2^{-(l+1)} \{t(\xi_n^{(2s_n-l)}) t^{(l)}(\xi_n^{(2s_n)}) - \Delta_\eta^{(K)}(\xi_n^{(2s_n-l)}) t^{(l-1)}(\xi_n^{(2s_n)})\}}{\prod_{k=0}^{l} d(\xi_n^{(2s_n-k)})}.
\end{aligned} \tag{5.17}
$$

Then, using (2.23) at the point $\lambda = \xi_n^{(2s_n)}$, taking into account the fact that $\xi_n^{(2s_n-l)} = \xi_n^{(2s_n)} + l\eta$, we get:

$$
t(\xi_n^{(2s_n-l)}) t^{(l)}(\xi_n^{(2s_n)}) - \Delta_\eta^{(K)}(\xi_n^{(2s_n-l)}) t^{(l-1)}(\xi_n^{(2s_n)}) = t^{(l+1)}(\xi_n^{(2s_n)}), \tag{5.18}
$$

hence leading to the formula:

$$Q_{t,n}^{(2s_n-l-1)} = \frac{k_2^{-(l+1)}t^{(l+1)}(\xi_n^{(2s_n)})}{\prod_{k=0}^{l} d(\xi_n^{(2s_n-k)})},\qquad(5.19)$$

which proves the induction hypothesis and hence (5.16) for any $h_n \in \{1,...,2s_n-1\}$.

Therefore, the existence of the polynomial $Q_t(\lambda)$ of the form (5.1) which satisfies with $t(\lambda)$ the quantum spectral curve equation is reduced to the proof of the existence of a polynomial of maximal degree $N_s$ that interpolates the values (5.13) in the $Q_t(\xi_n^{(h_n)})$ for any $h_n \in \{0,...,2s_n\}$ and $n \in \{1,...,N\}$. From the form (5.1), we can use the following interpolation formula in the $N_s$ points $\xi_n^{(h_n)}$ for $h_n \in \{1,...,2s_n\}$ and $n \in \{1,...,N\}$ and in an additional point $\zeta$:

$$Q_t(\lambda) = \sum_{a=1}^{N}\sum_{h_a=1}^{2s_a} \frac{\lambda-\zeta}{\xi_a^{(h_a)}-\zeta} \prod_{\substack{b=1 \\ (b,k_b)\neq(a,h_a)}}^{N}\prod_{k_b=1}^{2s_b} \frac{\lambda-\xi_b^{(k_b)}}{\xi_a^{(h_a)}-\xi_b^{(k_b)}}Q_t(\xi_a^{(h_a)}) + \prod_{b=1}^{N}\prod_{k_b=1}^{2s_b}\frac{\lambda-\xi_b^{(k_b)}}{\zeta-\xi_b^{(k_b)}}Q_t(\zeta),\quad(5.20)$$

where $\zeta$ is an arbitrary value different of $\xi_n^{(h_n)}$ for any $h_n \in \{1,...,2s_n\}$ and $n \in \{1,...,N\}$. Imposing now that $Q_t(\lambda)$ satisfies (5.13) for any $h_n \in \{1,...,2s_n\}$ and $n \in \{1,...,N\}$, we get:

$$Q_t(\lambda) = \sum_{a=1}^{N} \mathsf{q}_{t,a}\left(\sum_{h_a=1}^{2s_a} \frac{\lambda-\zeta}{\xi_a^{(h_a)}-\zeta} \prod_{\substack{b=1 \\ (b,k_b)\neq(a,h_a)}}^{N}\prod_{k_b=1}^{2s_b} \frac{\lambda-\xi_b^{(k_b)}}{\xi_a^{(h_a)}-\xi_b^{(k_b)}}Q_{t,a}^{(h_a)}\right) + \prod_{b=1}^{N}\prod_{k_b=1}^{2s_b}\frac{\lambda-\xi_b^{(k_b)}}{\zeta-\xi_b^{(k_b)}}\mathsf{q}_{t,0},\quad(5.21)$$

where we have introduced the notations:

$$\mathsf{q}_{t,0} \equiv Q_t(\zeta),\quad \mathsf{q}_{t,a} \equiv Q_t(\xi_a^{(2s_a)})\ \forall a \in \{1,...,N\},\qquad(5.22)$$

and where the values $Q_{t,a}^{(h_a)}$ are given in terms of the transfer matrix eigenvalues by (5.16). Hence, the polynomial $Q_t(\lambda)$ constructed from such an interpolation formula indeed takes the values $Q_t(\xi_a^{(h_a)})$ in the points $\lambda = \xi_a^{(h_a)}$ for $h_a \in \{1,...,2s_a\}$ and $a \in \{1,...,N\}$. Therefore what remains to be proven is that there exits a choice of the values $\mathsf{q}_{t,a}$, for $a \in \{1,...,N\}$, such that the polynomial $Q_t(\lambda)$ constructed by (5.21) indeed takes the values $Q_t(\xi_n^{(0)})$ in the points $\xi_n^{(0)}$ that were not used in the interpolation formula (5.21). So imposing these conditions indeed constitutes $N$ constraints on the possible values of the $\mathsf{q}_{t,a}$, for $a \in \{1,...,N\}$. So that we are left with the system of $N$ equations obtained by imposing that the interpolation formula (5.21) indeed satisfies the equations (5.13) in the points $h_n = 0$ for $n \in \{1,...,N\}$. This is an homogeneous linear system of $N$ equations in $N+1$ unknowns, the $\mathsf{q}_{t,a}$ for any $a \in \{0,...,N\}$, or equivalently an inhomogeneous system of $N$ equations in the $N$ unknowns, the $\mathsf{q}_{t,a}$ for any $a \in \{1,...,N\}$ in terms of the normalization $\mathsf{q}_{t,0}$:

$$\sum_{b=1}^{N}[C_\zeta^{(t)}]_{ab}\,\mathsf{q}_{t,b} = -\prod_{c=1}^{N}\prod_{k_c=1}^{2s_c}\frac{\xi_a^{(0)}-\xi_c^{(k_c)}}{\zeta-\xi_c^{(k_c)}}\mathsf{q}_{t,0},\qquad(5.23)$$

where the $N \times N$ matrix $[C_\zeta^{(t)}]_{ab}$ has the following elements

$$[C_\zeta^{(t)}]_{ab} = -\delta_{ab}\,Q_{t,a}^{(0)} + \sum_{h_b=1}^{2s_b}\frac{\xi_a^{(0)}-\zeta}{\xi_b^{(h_b)}-\zeta}\prod_{\substack{c=1 \\ (c,k_c)\neq(b,h_b)}}^{N}\prod_{k_c=1}^{2s_c}\frac{\xi_a^{(0)}-\xi_c^{(k_c)}}{\xi_b^{(h_b)}-\xi_c^{(k_c)}}Q_{t,b}^{(h_b)},\qquad \forall a,b \in \{1,\dots,N\},$$

$$(5.24)$$

where the coefficients $Q_{t,b}^{(h_b)}$ with $h_b \in \{0, \dots, 2s_b\}$, for any $b \in \{1, \dots, N\}$, are given from the transfer matrix eigenvalues by (5.16) as the solution to the linear system (5.11). The linear system (5.23) admits always one nonzero solution which produces one polynomial $Q_t(\lambda)$ satisfying the functional equation (5.2) with $t(\lambda)$. This and the uniqueness of the polynomial solution implies that $\det_N[C_\zeta^{(t)}]$ is nonzero and finite for almost any choice of $\zeta$. Then, for any given choice of $q_{t,0} \neq 0$, there exists one and only one nontrivial solution $(q_{t,1}, \dots, q_{t,N})$ of the system (5.23), which is given by Cramer's rule:

$$q_{t,j} = q_{t,0} \frac{\det_N[C_\zeta^{(t)}(j)]}{\det_N[C_\zeta^{(t)}]}, \qquad \forall j \in \{1, \dots, N\}, \tag{5.25}$$

with matrices $C_\zeta^{(t)}(j)$ defined by

$$[C_\zeta^{(t)}(j)]_{ab} = (1 - \delta_{b,j})[C_\zeta^{(t)}]_{ab} - \delta_{b,j} \prod_{b=1}^{N} \prod_{k_b=1}^{2s_b} \frac{\xi_a^{(0)} - \xi_b^{(k_b)}}{\zeta - \xi_b^{(k_b)}}, \tag{5.26}$$

for all $a, b \in \{1, \dots, N\}$. Let us now prove that the following conditions hold:

$$\det_N[C_\zeta^{(t)}(j)] \neq 0 \ \forall j \in \{1, \dots, N\}, \tag{5.27}$$

for almost any values of the parameters. Let us consider the case $k_1 = 0$ then the transfer matrix reduces to $k_2 D(\lambda)$, which coincides with $B^{(K)}(\lambda)$ for $K = k_2 \sigma_1$, whose diagonalizability and spectrum simplicity follows for example by Theorem 3.2. The spectrum explicitly reads:

$$t_{\mathbf{h}}^{(k_1=0, k_2 \neq 0)}(\lambda) \equiv k_2 \prod_{n=1}^{N} (\lambda - \xi_n^{(h_n)}) \ \forall \mathbf{h} \equiv \{h_1, \dots, h_N\} \in \bigotimes_{n=1}^{N} \{0, \dots, 2s_n\}. \tag{5.28}$$

In this special case we can solve explicitly in the class of the polynomial solutions the quantum spectral curve equation which reads:

$$t(\lambda)Q_t(\lambda) - k_2 d(\lambda)Q_t(\lambda + \eta) = 0. \tag{5.29}$$

Indeed, the polynomial

$$Q_{t_{\mathbf{h}}^{(k_1=0, k_2 \neq 0)}}(\lambda) = \prod_{n=1}^{N} \prod_{k_n=1}^{h_n - 1} (\lambda - \xi_n^{(k_n)}), \tag{5.30}$$

it is easily checked to satisfy the above functional equation (5.29) with $t_{\mathbf{h}}^{(k_1=0, k_2 \neq 0)}(\lambda)$ for any fixed $\mathbf{h} \in \bigotimes_{n=1}^{N} \{0, \dots, 2s_n\}$. Moreover, it is the only solution to the above equation for any fixed $\mathbf{h}$. This follows by the general proof of uniqueness, given above by the quantum Wronskian argument, or by rederiving it directly in this special case. Indeed, any polynomial solution of (5.29) has to have the factorized form:

$$\bar{Q}_{t_{\mathbf{h}}^{(k_1=0, k_2 \neq 0)}}(\lambda) = P(\lambda)Q_{t_{\mathbf{h}}^{(k_1=0, k_2 \neq 0)}}(\lambda), \tag{5.31}$$

for some polynomial $P(\lambda)$ which has to satisfy then the equation:

$$P(\lambda) - P(\lambda + \eta) = 0, \tag{5.32}$$

which is only possible for $P(\lambda)$ constant. Let us observe that all our polynomials $Q_{t_{\mathbf{h}}^{(k_1=0, k_2 \neq 0)}}(\lambda)$ are indeed of degree $M \leq N_s \equiv 2 \sum_{n=1}^{N} s_n$, so that for any fixed $\mathbf{h}$ we can interpolate them

in $N_s + 1$ points according to the interpolation formula (5.20). Moreover, as the couple $t_{\mathbf{h}}^{(k_1=0,k_2\neq 0)}(\lambda)$ and $Q_{t_{\mathbf{h}}^{(k_1=0,k_2\neq 0)}}(\lambda)$ satisfies the functional equation (5.29) then they satisfy also the homogeneous system of equations (5.11) or equivalently the system (5.13) with (5.14) and (5.15), for $k_1 = 0$. This means that the interpolation formula (5.21) also holds for our polynomial $Q_{t_{\mathbf{h}}^{(k_1=0,k_2\neq 0)}}(\lambda)$ and so imposing the condition (5.13) for any $h_a = 0$, we get the linear system (5.23), which is satisfied for:

$$\mathsf{q}_{t_{\mathbf{h}}^{(k_1=0,k_2\neq 0)},0} \equiv Q_{t_{\mathbf{h}}^{(k_1=0,k_2\neq 0)}}(\zeta) \neq 0, \quad \mathsf{q}_{t_{\mathbf{h}}^{(k_1=0,k_2\neq 0)},a} \equiv Q_{t_{\mathbf{h}}^{(k_1=0,k_2\neq 0)}}(\xi_a^{(2s_a)}) \neq 0 \ \forall a \in \{1,...,N\}. \tag{5.33}$$

This in turn implies that (5.27) is satisfied for $k_1 = 0, k_2 \neq 0$, being $\det_N[C_\zeta^{(t)}]$ nonzero by the uniqueness of the polynomial solution. Now we have just to observe that the transfer matrix eigenvalues are algebraic functions in the parameter $k_1$ so the same is true for the determinants $\det_N[C_\zeta^{(t)}]$ and $\det_N[C_\zeta^{(t)}(j)]$ for any $j \in \{1,...,N\}$. Then, by using the Lemma B.1 of our article [2], we can argue that being these determinant nonzero at $k_1 = 0$ for any transfer matrix eigenvalue this implies that this must be true for almost any values of the parameters.

Finally, let us note that $Q_t(\lambda)$ being a non zero (by construction) polynomial of maximal know degree, its highest coefficient can always be normalized to unity as required. □

## 5.2 Reconstruction of the $Q$-operator

On the basis of the results derived in the SoV framework we can present a reconstruction of the $Q$-operator in terms of the (fused) transfer matrices, more precisely it holds:

**Corollary 5.1.** *Let us assume that the twist matrix $K \in \mathrm{End}(\mathbb{C}^2)$ is such that $k_1 \neq k_2$, $k_1 \neq 0$, $k_2 \neq 0$. Then the polynomial family of commuting operators of maximal degree $N_s$ constructed from the fused transfer matrices:*

$$Q(\lambda) = \frac{\det_N[C_\zeta^{(T^{(K)})} + \Upsilon_\zeta^{(T^{(K)})}(\lambda)]}{\det_N[C_\zeta^{(T^{(K)})}]} \prod_{b=1}^{N} \prod_{k=1}^{2s_b} \frac{\lambda - \xi_b^{(k)}}{\zeta - \xi_b^{(k)}}, \tag{5.34}$$

*is well defined for any fixed value[8] of $\zeta$ and for almost any values of $\{\xi_{i\leq N}\}$ and of $\{k_{j\leq 2}\}$, where we have defined:*

$$[C_\zeta^{(T^{(K)})}]_{ab} = -\delta_{ab} Q_{T^{(K)},a}^{(0)} + \sum_{h=1}^{2s_b} \frac{\xi_a^{(0)} - \zeta}{\xi_b^{(h)} - \zeta} \prod_{\substack{c=1 \\ (c,k)\neq(b,h)}}^{N} \prod_{k=1}^{2s_c} \frac{\xi_a^{(0)} - \xi_c^{(k)}}{\xi_b^{(h)} - \xi_c^{(k)}} Q_{T^{(K)},b}^{(h)} \qquad \forall a,b \in \{1,\dots,N\}, \tag{5.35}$$

*with:*

$$Q_{T^{(K)},a}^{(h)} = \frac{T^{(K|2s_a-h)}(\xi_a^{(2s_a)})}{k_2^{(2s_a-h)} \prod_{k=0}^{2s_a-(h+1)} d(\xi_a^{(2s_a-k)})}, \qquad \forall h \in \{1,\dots,2s_a\}, a \in \{1,\dots,N\}, \tag{5.36}$$

---

[8]We can fix for example $\zeta = \xi_a^{(0)}$ for any fixed $a \in \{1,\dots,N\}$.

and $\Upsilon_\zeta^{(\mathsf{T}^{(K)})}(\lambda)$ is the following rank one matrix:

$$
[\Upsilon_\zeta^{(\mathsf{T}^{(K)})}(\lambda)]_{ab} = \prod_{c=1}^{\mathsf{N}}\prod_{k=1}^{2s_c}(\xi_a^{(0)} - \xi_c^{(k_c)})
$$
$$
\sum_{h=1}^{2s_b}\frac{(\lambda - \zeta)\,\mathsf{Q}_{\mathsf{T}^{(K)},b}^{(h)}}{(\xi_b^{(h)}-\lambda)(\xi_b^{(h)}-\zeta)\prod_{\substack{c=1\\(c,k_c)\neq(b,h)}}^{\mathsf{N}}\prod_{k=1}^{2s_b}(\xi_b^{(h)}-\xi_c^{(k_c)})}\quad \forall a,b\in\{1,\dots,\mathsf{N}\}.\quad (5.37)
$$

Then the family $\mathsf{Q}(\lambda)$ is a Q-operator family, namely it satisfies, together with the transfer matrix $\mathsf{T}^{(K)}$ and the quantum determinant $\Delta_\eta^{(K)}$, the quantum spectral curve equation at operator level:

$$
\alpha(\lambda)\mathsf{Q}(\lambda - 2\eta) - \beta(\lambda)\mathsf{T}^{(K)}(\lambda - \eta)\mathsf{Q}(\lambda - \eta) + \Delta_\eta^{(K)}(\lambda)\mathsf{Q}(\lambda) = 0, \qquad (5.38)
$$

and for any $a\in\{1,\dots,\mathsf{N}\}$ $\mathsf{Q}(\xi_a^{(2s_a)})$ are invertible operators.

*Proof.* We have shown in the previous theorem that a unique polynomial $Q_t(\lambda)$ of the form (5.1) satisfies the quantum spectral curve equation for any fixed $t(\lambda)$ eigenvalue of the transfer matrix $\mathsf{T}^{(K)}(\lambda)$. Then by using the reconstruction of $Q_t(\lambda)$ in the points $\xi_a^{(2s_a)}$ for any $a\in\{1,\dots,\mathsf{N}\}$ and its interpolation formula (5.21) it is easy to prove that the following determinant representation holds:

$$
Q_t(\lambda) = \frac{\det_{\mathsf{N}}[C_\zeta^{(t)} + \Upsilon_\zeta^{(t)}(\lambda)]}{\det_{\mathsf{N}}[C_\zeta^{(t)}]}\prod_{b=1}^{\mathsf{N}}\prod_{k=1}^{2s_b}\frac{\lambda - \xi_b^{(k)}}{\zeta - \xi_b^{(k)}}, \qquad (5.39)
$$

where we have replaced any transfer matrix $\mathsf{T}^{(K)}(\xi_a)$ by its eigenvalue $t(\xi_a)$ in the above defined matrices. Now as a corollary of Proposition 2.5 of our first paper [1], we know that the transfer matrix $\mathsf{T}^{(K)}(\lambda)$ is diagonalizable and with simple spectrum for almost any value of the parameters $\{\xi_{i\leq \mathsf{N}}^{(0)}\}$ and $\{\mathsf{k}_{j\leq 2}\}$ for the twist matrix $K$ diagonalizable and with simple spectrum. The polynomial operator family $\mathsf{Q}(\lambda)$ can be then uniquely defined by its action on the eigenbasis of the transfer matrix as it follows:

$$
\mathsf{Q}(\lambda)|t\rangle = |t\rangle Q_t(\lambda), \qquad (5.40)
$$

for any $t(\lambda)$ eigenvalue and uniquely (up to normalization) associated eigenvector $|t\rangle$ of the transfer matrix $\mathsf{T}^{(K)}(\lambda)$. It is then evident that this operator family satisfies by definition the quantum spectral curve equation with the transfer matrices, that it admits the given determinant representation in terms of the transfer matrix $\mathsf{T}^{(K)}(\lambda)$ and that it is invertible in the points $\{\xi_{a\leq \mathsf{N}}^{(2s_a)}\}$. $\qquad\square$

## 5.3 On the general role of the Q-operator as SoV basis generator

The known results on SoV in the literature and, in particular, our general construction of the SoV bases allows to show that for a large class of integrable quantum models associated to finite dimensional representation of the Yang-Baxter, reflection algebra or dynamical generalization of them, the transfer matrix (or some simple extension of it) defines a diagonalizable and simple spectrum one parameter family of commuting operators. Moreover, for the same class of models we know for a fixed normalization of the eigenvectors that the corresponding wave functions admit the following factorized form:

$$
\langle h_1,\dots,h_N|t\rangle = \prod_{a=1}^{\mathsf{N}}Q_t(\xi_a^{(h_a)}),\quad \forall h_a\in\{1,\dots,d_a\}, a\in\{1,\dots,\mathsf{N}\}\qquad (5.41)
$$

for a quantum space of dimension $\prod_{a=1}^{N} d_a$, where $Q_t(\lambda)$ is the eigenvalue of the $Q$-operator $Q(\lambda)$ associated to the given transfer matrix eigenvalue and the $\xi_a^{(h_a)}$ is the spectrum of the quantum separate variables $Y_a$ satisfying:

$$\langle h_1,...,h_N | Y_a = \xi_a^{(h_a)} \langle h_1,...,h_N |, \quad \forall a \in \{1,\ldots,N\}. \tag{5.42}$$

Once we combine the diagonalizability and simplicity of the transfer matrix spectrum and the SoV representation of the transfer matrix eigenco-vectors then it is clear that the following statement holds:

There exists a co-vector $\langle L |$ such that:

$$\langle h_1,...,h_N | = \langle L | \prod_{a=1}^{N} Q(\xi_a^{(h_a)}), \quad \forall h_a \in \{1,\ldots,d_a\}, a \in \{1,\ldots,N\} \tag{5.43}$$

indeed:

$$\langle L | \prod_{a=1}^{N} Q(\xi_a^{(h_a)}) | t \rangle = \prod_{a=1}^{N} Q_t(\xi_a^{(h_a)}) \langle L | t \rangle, \quad \forall h_a \in \{1,\ldots,d_a\}, a \in \{1,\ldots,N\}, \tag{5.44}$$

clearly the definition of $\langle L |$ is fixed up to the choice of the normalization of all the transfer matrix eigenvectors. These observations naturally lead to the idea that we are able to construct an SoV basis once the $Q$-operator is known. There is anyhow an important comments we have to make, i.e. some further informations are indeed required beyond the knowledge of the $Q$-operator. In particular, the right choice of the vector $\langle L |$ which has to satisfy the condition:

$$\langle L | t \rangle \neq 0 \tag{5.45}$$

for $|t\rangle$ any transfer matrix eigenvector and even more importantly we have to have a criterion to chose the spectrum of the separate variables. In our SoV construction there are indeed the fusion relations and the fact that they are simplified for some specific choice of the values of the spectral parameter to guide us to the proper choice of the spectrum of the quantum separate variables.

In the non-fundamental representations we considered in this article, we can now make the above description of the SoV basis construction using the $Q$-operator. In particular, we have the following:

**Corollary 5.2.** *Let us assume that the twist matrix $K \in \text{End}(\mathbb{C}^2)$ is such that $k_1 \neq k_2$, $k_1 \neq 0$, $k_2 \neq 0$, then for almost any choice of $\langle L |$ the set of co-vectors defined in (5.43) defines an SoV co-vector basis which coincides, thanks to equations 5.13 and 5.16, with that defined in Theorem 4.2, once we fix:*

$$\langle L | = \langle S | \prod_{n=1}^{N} Q^{-1}(\xi_n^{(2s_n)}). \tag{5.46}$$

*Moreover, the set of co-vectors (5.43) coincides with Sklyanin's SoV basis under the choices of $\langle S |$ described in Theorem 4.2.*

Let us look now at the linear action of the transfer matrix on the basis (5.43) generated by the $Q$-operator. It follows directly from the $T - Q$ relation that gives the product $T^{(K)}(\lambda)Q(\lambda)$ as a linear combination of $Q(\lambda + \eta)$ and $Q(\lambda - \eta)$, more precisely we get:

$$T^{(K)}(\xi_a^{(h_a)})Q(\xi_a^{(h_a)}) = k_1 a(\xi_n^{(h_n)})Q(\xi_a^{(h_a)} - \eta) + k_2 d(\xi_n^{(h_n)})Q(\xi_a^{(h_a)} + \eta). \tag{5.47}$$

Now taking into account $\xi_a^{(h_a)} \mp \eta = \xi_a^{(h_a \pm 1)}$ it leads directly to the following explicit linear action on the SoV basis (5.43) as:

$$\langle h_1, ..., h_N | \mathsf{T}^{(K)}(\xi_n^{(h_n)}) = \mathsf{k}_1 a(\xi_n^{(h_n)}) \langle h_1, ..., h_n + 1, ..., h_N | + \mathsf{k}_2 d(\xi_n^{(h_n)}) \langle h_1, ..., h_n - 1, ..., h_N |, \tag{5.48}$$

which coincides with (4.35). It should be noted here that whenever the index $h_n$ reaches the value 0 or $2s_n$ the corresponding coefficient vanishes ensuring that the action never goes out of the basis. Namely it can be easily verified that $a(\xi_n^{(2s_n)}) = 0$ and similarly $d(\xi_n^{(0)}) = 0$. It is then sufficient to get the explicit action of $\mathsf{T}^{(K)}(\lambda)$ to use its expression following from the interpolation formula for the operator $\mathsf{T}^{(K)}(\lambda)$ in the points $\xi_n^{(h_n)}$ for $n = 1, ..., N$. Hence the role of the $T-Q$ equation is both to give the $T$-spectrum characterization and at the same time the closure relation leading to its explicit linear action on the SoV basis generated by the $Q$-operator. This is very similar to the Frobenius scheme that we advocated in [1], here with the $T-Q$-equation playing the natural role of the characteristic equation for the transfer matrix.

A last remark about the construction of the SoV basis starting from the $Q$-operator should be outlined. For several integrable quantum models it is in fact the construction of the SoV basis and the corresponding characterization of the transfer matrix spectrum that allow for the explicit construction of the $Q$-operator, as we have explained in this article.

# 6 Conclusion

In the present paper we have shown how to construct different SoV bases for the integrable models associated to higher spin representations of the $Y(gl_2)$ algebra. Here, we would like to comment on some general picture relating these SoV bases. In our scheme developed first in [1] an SoV basis of such a model is characterized by a chosen reference co-vector $\langle L|$, satisfying the condition (5.45), and a set of commuting conserved charges $\mathsf{T}_i^{(h_i)}$, with $i = 1, ..., N$ and $h_i = 0, ..., 2s_i$, such that the set of co-vectors:

$$\langle L| \prod_{i=1}^{N} \mathsf{T}_i^{(h_i)} \equiv {}_\mathsf{T}\langle h_1, ..., h_N| \equiv {}_\mathsf{T}\langle \mathbf{h}|, \tag{6.1}$$

forms a basis of the Hilbert space dual $\mathcal{H}^*$ of the model at hand. Moreover, the conserved charges are obtained from some transfer matrix of the model, stemming from the Yang-Baxter algebra, and satisfy a set of low-degree algebraic equations determining their common spectrum. In the cases examined in this paper, these are given by the fusion relations for the tower of fused transfer matrices and define a system of $N$ polynomial equations of order $2s + 1$ in $N$ unknowns, if all $s_i = s$. The low degree of these algebraic equations determining the spectrum of the transfer matrix (and also its action on the constructed SoV basis) in comparison to the dimension of the Hilbert space $((2s + 1)^N$ in the case here considered) is the hallmark of integrability. Indeed, in general, the degree of the characteristic polynomial of an operator is equal to the dimension of the Hilbert space; the very fact that there exists lower degree equations giving the spectrum is directly related to the Yang-Baxter algebra structure underlying the integrability properties of the considered models.

Now, suppose we have an SoV basis as described above; what freedom do we have for constructing different SoV bases? There are two obvious ways to do so:

(i) Change the reference co-vector $\langle L|$ for another co-vector, say $\langle \tilde{L}|$.

(ii) Change the set of commuting conserved charges $\mathsf{T}_i^{(h_i)}$ to a new set, say $\tilde{\mathsf{T}}_i^{(k_i)}$.

Let us stress that in the second case, while changing the set of commuting conserved charges, we need to insure that again the spectrum of the transfer matrix is given by low degree (in the above described way) algebraic equations that also play the role of closure relations for the action of the transfer matrix on the new basis. In the present paper we have been giving examples of the two above ways, and even of combined effects of the two. Before commenting directly on the examples given in the present paper, let us examine on general grounds these two possibilities more closely.

Let us first suppose that we have two different reference states $\langle L|$ and $\langle \tilde{L}|$ for which the set of co-vectors (6.1) forms a basis of $\mathcal{H}^*$ and such that the set of co-vectors defined by:

$$\langle \tilde{L}| \prod_{i=1}^{N} \mathsf{T}_i^{(h_i)}, \tag{6.2}$$

is also a basis of $\mathcal{H}^*$. Then we can infer that the change of basis between the two sets is given by the action of an invertible operator, say $\mathsf{T}_{L,\tilde{L}}$, that is a conserved charge commuting with the transfer matrix. Indeed, (6.1) being a basis, there exist coefficients $\tilde{l}_{\mathbf{h}}$ such that:

$$\langle \tilde{L}| = \sum_{\mathbf{h}}{}_{\mathsf{T}}\langle \mathbf{h}|\tilde{l}_{\mathbf{h}}. \tag{6.3}$$

Hence, defining the conserved charge:

$$\mathsf{T}_{L,\tilde{L}} = \sum_{\mathbf{h}} \tilde{l}_{\mathbf{h}} \mathsf{T}_{\mathbf{h}}, \tag{6.4}$$

with $\mathsf{T}_{\mathbf{h}} \equiv \prod_{i=1}^{N} \mathsf{T}_i^{(h_i)}$, we get:

$$\langle \tilde{L}| \prod_{i=1}^{N} \mathsf{T}_i^{(h_i)} = \langle L| \prod_{i=1}^{N} \mathsf{T}_i^{(h_i)} \cdot \mathsf{T}_{L,\tilde{L}}. \tag{6.5}$$

Hence any change of SoV basis obtained by changing the reference co-vector $\langle L|$, is generated by the action of some conserved charge commuting with the transfer matrix.

Let us remark, moreover, that the invertibility of the charge $\mathsf{T}_{L,\tilde{L}}$ is the necessary and sufficient condition to guarantee that $\langle \tilde{L}|$ satisfies the condition (5.45), once $\langle L|$ satisfies it, being

$$\langle \tilde{L}|t\rangle = \mathsf{t}_{L,\tilde{L}}\langle L|t\rangle, \tag{6.6}$$

where $\mathsf{t}_{L,\tilde{L}}$ is the eigenvalue of the charge $\mathsf{T}_{L,\tilde{L}}$ on the generic transfer matrix eigenvector $|t\rangle$.

For both the type of SoV bases here introduced, we have been able to identify explicitly both reference co-vectors satisfying (6.3) and the set of charges $\mathsf{T}_i^{(h_i)}$. Then, we can explicitly write the condition on $\langle \tilde{L}|$ to be a proper reference co-vector as a condition on its decomposition in these explicit SoV bases. That is, it must holds:

$$\det\left[\sum_{\mathbf{h}} \tilde{l}_{\mathbf{h}}^{(1)} \prod_{n=1}^{N} (\mathsf{T}^{(K|2s_n)}(\xi_n^{(2s_n-1)}))^{h_n}\right] \neq 0, \tag{6.7}$$

where $\tilde{l}_{\mathbf{h}}^{(1)}$ are the coefficients defined in (6.3) taking as original SoV-basis the first one associated to the choice of $\langle \Omega|$ given in (4.2). Similarly, it must holds:

$$\det\left[\sum_{\mathbf{h}} \tilde{l}_{\mathbf{h}}^{(2)} \prod_{n=1}^{N} \frac{\mathsf{k}_2^{h_n-2s_n}\mathsf{T}^{(K|2s_n-h_n)}(\xi_n^{(2s_n)})}{\prod_{k=0}^{2s_n-h_n-1} d(\xi_n^{(2s_n-k)})}\right] \neq 0, \tag{6.8}$$

where $\tilde{l}_{\mathbf{h}}^{(2)}$ are the coefficients defined in (6.3) taking as original SoV-basis the second one associated to the choice of $\langle S|$ given in (4.19) or (4.21). Then, in agreement with our finding in Proposition (4.1) and in Theorem (4.2), we confirm that almost any choice of the co-vector $\langle \tilde{L}|$ defines a proper reference co-vector satisfying the condition (5.45).

Let us now consider the second procedure, namely let us suppose we have two SoV bases sharing the same reference co-vector $\langle L|$ but associated to two different sets of commuting conserved charges, $\mathsf{T}_{\mathbf{h}}$ and $\tilde{\mathsf{T}}_{\mathbf{k}}$. Then it means that each of these sets of commuting conserved charges are both bases of the vector space $\mathcal{C}_{\mathsf{T}}$ of operators commuting with the transfer matrix $\mathsf{T}(\lambda)$ of the model considered. Due to simple spectrum property of the transfer matrix, that follows from the existence of such an SoV basis, the dimension of the vector space $\mathcal{C}_{\mathsf{T}}$ is equal to the dimension of the Hilbert space on which these operators are acting upon. Hence, there exists a matrix $M_{\mathbf{h},\mathbf{k}}$ defining the corresponding change of basis such that:

$$\tilde{\mathsf{T}}_{\mathbf{k}} = \sum_{\mathbf{h}} \mathsf{T}_{\mathbf{h}} M_{\mathbf{h},\mathbf{k}}, \tag{6.9}$$

which also gives the change of basis in $\mathcal{H}^*$ as we also have:

$$\langle L|\tilde{\mathsf{T}}_{\mathbf{k}} \equiv {}_{\tilde{\mathsf{T}}}\langle \mathbf{k}| = \sum_{\mathbf{h}} {}_{\mathsf{T}}\langle \mathbf{h}| M_{\mathbf{h},\mathbf{k}}. \tag{6.10}$$

Note that the change of basis matrix $M_{\mathbf{h},\mathbf{k}}$ also gives the linear decomposition of the commuting conserved charges $\tilde{\mathsf{T}}_{\mathbf{k}}$ on the first set $\mathsf{T}_{\mathbf{h}}$. In that respect, one cannot pickup an arbitrary invertible matrix $M_{\mathbf{h},\mathbf{k}}$ and expect that it will be defining a new SoV basis; indeed, the size of this matrix being the dimension of the Hilbert space, one should further insure that the new set of commuting conserved charges generated by the formulae (6.10) still satisfy a system of algebraic equations of low degree in the above defined sense generating their spectrum, namely involving a number of terms much less than the dimension of the Hilbert space.

Finally, let us stress that in practice, we can have both effects (i) and (ii) when going from one SoV basis to another one. In fact one example of this combined effect is our second construction of the SoV basis for arbitrary spin representations given respectively in (4.17) and (4.18). In that case the two bases are just identical via the change of both the set of conserved charges used and also a change of the reference co-vector. The change of basis between our first construction (4.1), using the fundamental transfer matrices and the second construction using either the tower of the fused transfer matrices or the $Q$-operator, is more involved, but both construction can be obtained through determinant formulae giving the fused transfer matrices in terms of the original transfer matrix $\mathsf{T}^{(K|1)}(\lambda)$, and hence can be related using the fusion relations. The main advantage of our second construction over our first construction is that the action of the transfer matrix $\mathsf{T}^{(K|1)}(\lambda)$ becomes very simple, namely explicitly linear, to be compared to its action on the first SoV basis (4.1) which is more involved. In fact it is only in the second SoV basis that the action of the original transfer matrix $\mathsf{T}^{(K|1)}(\lambda)$ on the SoV basis is given by the same algebraic relation that also characterizes its spectrum, i.e., by the $TQ$-equation. Hence, it appears clearly that the most "natural" SoV bases are the ones for which this situation is realized. And for models associated to higher-spin representations of the $Y(gl_2)$ quantum algebra, we have shown in (5.43) that they are obtained through commuting conserved charges generated by the Baxter $Q$-operator, here, explicitly constructed in terms of the fused transfer matrices in Corollary 5.1.

## Acknowledgments

J. M. M. and G. N. are supported by CNRS and ENS de Lyon.

# A  Appendix

Let us define the following general tridiagonal matrix $\mathcal{T}_N^{(1,N)}$ :

$$\mathcal{T}_N^{(1,N)} = \begin{pmatrix} a_1 & b_1 & & 0 \\ c_1 & \ddots & \ddots & \\ & \ddots & \ddots & b_{N-1} \\ 0 & & c_{N-1} & a_N \end{pmatrix}, \tag{A.1}$$

and let us denote by $\Lambda_N^{(1,N)}$ its determinant. Expanding this determinant by its column $j$, we have:

$$\Lambda_N^{(1,N)} = a_j \Lambda_{j-1}^{(1,j-1)} \Lambda_{N-j}^{(j+1,N)} - b_{j-1}\Theta_{N-1}^{(1)} - c_j \Theta_{N-1}^{(2)}. \tag{A.2}$$

Then we can expand the two determinants $\Theta_{N-1}^{(1)}$ and $\Theta_{N-1}^{(1)}$ respectively along their rows $j-1$ and $j$ respectively, to get in both cases a sum of two contributions, one being zero and the other being a product of two determinants of tridiagonal matrices of the above form to get:

$$\Lambda_N^{(1,N)} = a_j \Lambda_{j-1}^{(1,j-1)} \Lambda_{N-j}^{(j+1,N)} - b_{j-1}c_{j-1}\Lambda_{j-2}^{(1,j-2)}\Lambda_{N-j}^{(j+1,N)} - b_j c_j \Lambda_{j-1}^{(1,j-1)}\Lambda_{N-j-1}^{(j+2,N)}, \tag{A.3}$$

hence leading to a relation expressing the determinant of the tridiagonal matrix $\mathcal{T}_N^{(1,N)}$, namely $\Lambda_N^{(1,N)}$, in terms of determinants of some of its sub-tridiagonal matrices. Taking the particular case where $j = N$, and with the obvious conventions that the coefficients $b_j$ and $c_j$, for $j \leq 0$ or for $j \geq N$ are identically zero, while the corresponding $a_j$ are equal to one if $j \leq 0$ or $j > N$, we get back to the traditional recursion relation for determinants of tridiagonal matrices:

$$\Lambda_N^{(1,N)} = a_N \Lambda_{N-1}^{(1,N-1)} - b_{N-1}c_{N-1}\Lambda_{N-2}^{(1,N-2)}. \tag{A.4}$$

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
