# Peer review of "On quantum separation of variables beyond fundamental representations"

_SciPost Physics, doi:SciPost Phys. 10, 026 (2021)_

## Round 1 · Referee Report · Jules Lamers · 2019-6-19

Strengths
1- Interesting results: extension of authors' new SoV approach beyond fundamental rep, with a 'natural' SoV basis on which the transfer matrix acts in a simple way; contact made with Sklyanin's SoV and Baxter's $TQ$ equation; explicit construction of $Q$-operator
2- Contains review of the necessary background, so essentially self contained (except for relying just a few times on some proofs in, I think, especially ref [1] from the submission)
3- Quite clearly structured
Weaknesses
1- English grammar
Report
In 2018 the authors found a new approach to Sklyanin's separation of variables (SoV) within the framework of the quantum-inverse scattering method. The present paper, which is already the fourth in this series of works, deals with quantum-integrable lattice models that go beyond the fundamental representation: for simplicity considering the case of a rational R-matrix and rank one, the authors extend their new approach to models with higher-dimensional representations (obtained by fusion) associated to the lattice sites, where the dimensions $2s_n +1$ ($1\leq n\leq N$) may vary from site to site. To ensure a simple spectrum the inhomogeneities and twisted boundaries need to be generic to ensure a simple spectrum.
If the auxiliary space is $V_a \cong \mathbb{C}^2$ let's call $A+D$ the 'auxiliary transfer matrix' (cf the notation in ref [12] of the submission), and use 'fundamental transfer matrix' for the case $V_a \cong \mathbb{C}^{2s_n+1}$ for some $1\leq n\leq N$.
The authors review Sklyanin's SoV when the twist is not proportional to the identity, so that its off-diagonal entries can be made nonzero by $\mathfrak{gl}_2$-invariance. Then the repeated action of the $A$-operator (evaluated at paticular shifts of the inhomogeneities) gives an explicit construction of a (left) $B$-eigenbasis on which $A$ and $D$ have a simple action.
Next, following the authors' work for fundamental models, a SoV covector basis of the Hilbert space is constructed by the repeated application of fundamental transfer matrices (evaluated at the same shifts of inhomogeneities). In this setting the spectrum of the auxiliary transfer matrix can be characterised ('SoV discrete characterisation') by $N$ polynomials determined in terms of solutions to a system of polynomial equations (for the spectrum at those same shifts of inhomogeneities). Moreover, for each such polynomial there exists (up to normalisation) a unique (left) eigenvector of the auxiliary transfer matrix with factorised components with respect to the SoV covector basis.
In addition the authors construct a new, second ('natural') SoV basis, where the $h_n$-fold action of the fundamental transfer matrix (with $V_a \cong \mathbb{C}^{2s_n+1}$) is replaced by something simpler: a single application of the fused transfer matrix (evaluated at the same value) with $V_a \cong \mathbb{C}^{2s_n - h_n +1}$ of intermediate dimension. (One can't help but think here of the work of Ilievski et al on quasilocal charges, where the tower of fused transfer matrices also play an important role.) The relation of this construction with Sklyanin's SoV covectors is given. The auxiliary transfer matrix has a simple action (as is easily seen from the fusion relation). As before its eigenvalues has a 'discrete' SoV characterisation in terms of a set of polynomials depending on solutions to a system of polynomial equations; and the corresponding eigenvectors have factorised components with respect to this covector basis too.
The authors go on to reformulate their preceding 'discrete characterisation' of the auxiliary transfer-matrix spectrum in a functional equation of Baxter $TQ$-type (the 'quantum spectral curve'). The factorisation of the components of the eigenvectors features the same polynomials $Q$. Furthermore, the corresponding $Q$-operator is explicitly constructed in terms of fused transfer matrices. Finally the natural SoV covector basis is given in terms of this $Q$-operator.
* * *
These results are certainly very interesting. The proofs look correct. (I have not gotten around to check the proof of Theorem 5.1, or those from ref [1] in the submission that are used, in detail.)
My main point of criticism is just that the presentation would benefit from improved grammar (with some concrete examples below), perhaps with the help of feedback from a native speaker, to help the reader focus on the contents. Below I give more detailed comments.
Requested changes
1- Abstract: "...two different SoV basis" -> "...two different SoV bases".
2- Here "the Sklyanin's approach" looks ungrammatical to me; e.g. -> "the approach of Sklyanin" or -> "Sklyanin's approach". Likewise for various similar phrases elsewhere.
3- (Optional) p1: perhaps "secular equation" -> "characteristic equation" is more standard?
4- p4: "$Y(gl_{n\geq 2})$" looks somewhat sloppy to me; I'd prefer "$Y(gl_n)$, $n\geq 2$". Likewise for similar notation elsewhere.
5- p5: "bi-dimensional" -> "two-dimensional" (and I believe there are several occurrences of this)
6- same paragraph: to distinguish between the different transfer matrices perhaps consider introducing some name also for the non-fundamental case (which I chose to call 'auxiliary' above)
7- p6: Is the phrase "associative and commutative algebra" used for a particular reason? Something like "commutative (associative) algebra" sounds more natural to me.
8- below (2.1): "spin-$s$" -> "higher-spin" or "spin-$s_n$"
9- below (2.6): "...of rational..." -> "...of the rational..."
10- (2.7): for more uniform notation, cf (2.3), consider including a subscript 0 for the matrix
11- below (2.7): "where we have define ..." -> "where we have defined ..."
12- (2.9): perhaps superscript $K|1$ here? The current notation is really only introduced in (2.22)
13- (2.10) and elsewhere: the authors could consider using "\mathrm{qdet}" rather than the current subscript $q$, cf the (different) notation in e.g. (4.17)
14- below (2.10): "central elements" -> "central element"
15- below (2.15): "it is an" -> "is an"
16- (2.20): perhaps add "for all $\lambda, \mu$" too
17- (2.21) and elsewhere: to improve readability perhaps use parentheses around the argument of the quantum determinant
18- (2.23): mention that $k_1, k_2$ are the eigenvalues of $K$ (currently done on the top of the next page)
19- above (2.26) and elsewhere likewise: I would prefer writing out abreviations like "w.r.t."
20- (2.30): could avoid introducing $\theta$ (which I think is only used here?) by adding a few words.
21- perhaps move the sentence below (2.33) to just above (2.31)
22- below (3.1) and elsewhere: "it exists" -> "there exists"
23- bottom p10 and elsewhere: "zeros" -> "zeroes"
24- (3.5) and (3.6): is there an inverse missing on the second $W$? Also, spend at least a few words on the $\mathcal{W}$s; these are defined by (3.5)?
25- below (3.7): "opf" -> "of", "local left references states" -> "local left reference states"
26- Thm 3.2: "are verified" -> "is verified", "non proportional" -> "not proportional". Perhaps "..._{Sk}" -> "..._\text{Sk}" here and elsewhere.
27- above (3.23): "is $B^{(K)}$-eigencovector" -> "is a $B^{(K)}$-eigencovector"
28- above (3.27): "being it" -> "it being"
From now on I will only suggest grammatical corrections if I believe they would prevent unclarities.
29- Prop 4.1: Presently the underline has a different meaning for the Sklyanin SoV (Sect 3) and the SoV of the authors (Sect 4). I think this is somewhat confusing, so perhaps one of these underlines can be changed to some other notation.
30- (4.3): $h$ seems to be missing (as an exponent?)
31- above (4.4): "same lines that in" -> "same lines as in". (In fact, I believe "follows the same lines" is not the standard formulation; -> "follows" or "is/goes/proceeds/... along the lines of" or "is the same as" or "is similar to". This comment applies elsewhere too.)
32- (4.5), (4.6) and following sentence: Lax operator instead of $R$-matrix?
33- (4.9): I think the first "$t(\lambda):$" and the "$\forall$" can be removed?
34- below (4.9): "solutions to" -> "that solve" or "solving". (By the way, here and elsewhere: "$N$-uple" and "$N$-upla" -> "$N$-tuple".)
35- below (4.10): perhaps start sentence with "each of which is"
36- (4.12): is this $D_{t,n}$" with superscript the same as that without?
37- Am I correct that (4.10) is the SoV analogue of the BAE? If so, are they easier to solve? (The counting of their solutions clearly is.) How would they compare for the TBA?
38- Top of p17: reference should be in the references, not a footnote
39- Also, what is a 'component' of a polynomial?
40- below (4.15) and elsewhere: I think that the $\bigotimes$ should be a Cartesian product?
41- Thm 4.2: "The set" -> "The set consisting of"; after all, (4.19) is not a set
42- (4.19) and elsewhere: can the parentheses in the power of $k_2$ be dropped? Looks like a superscript now.
43- p19: current notation is a bit confusing to me; I don't think there's any need to have $\bar{h}$s, $h$s, as well as $\hat{h}$s. Can't we just assume the result holds for some $h_1,\cdots,h_N$ (or all with bars) and consider $h_1,\cdots,h_n-1,\cdots,h_N$ (then also with bars) for arbitary $1\leq n\leq N$, and then with $h_1,\cdots,2s_n,\cdots,h_N$ (or $h_n=2s_n$ if you prefer, and use bars elsewhere).
44- Top of p21: "invariant w.r.t." -> "independent of"
45- (4.41): perhaps good to repeat the meaning of the notation $t^{(2s_n-h_n)}$ (with the superscript) from the proof of Thm 4.1, see the line below (4.16).
46- (5.1) is a bit unclear to me: should the "such that ..." go perhaps to the next line, or "for any $(a,b)\in ...$" to the (4.5)?
47- (5.2) perhaps it would be useful to give the explicit form, using (2.11), too, as this might be more familiar to the reader (and is used in the proof)
48- below (5.13): "trigonal" -> "tridiagonal"
49- p25 and elsewhere: "unicity" -> "uniqueness"
50- above Cor 5.1: "monodromy matrix" -> "transfer matrix"?
51- Cor 5.1: I assume that the meaning of "it is a $Q$-operator family" is what's explained at the end of the corrolary?
52- Sect 5.3: the first sentence is a bit hard to parse at the moment.
53- Around (5.45): what are the precise conditions on $\langle L|$ and other assumptions needed to construct the SoV basis from the $Q$-operator?
54- (5.46): perhaps refer to (4.20) and/or write $\langle 2s_1 ,\cdots 2s_n|$ instead?
55- Final paragraph: is this statement new, or can it already be found in the literature?
56- If I'm not mistaken, cf ref [12] in the submission, in the ABA for higher spin the auxiliary transfer matrix gives the BAE, while the fundamental transfer matrix the Hamiltonian and energy. What about the ('discrete' and 'functional') SoV characterisation of the spectrum of the fundamental transfer matrix, especially in the case where all $s_n$ are equal to some fixed spin $s$?

---

## Round 1 · Referee Report · Anonymous · 2019-7-25

Strengths
1- extension of SoV approach to $Y(gl_2$) quantum spin chains with mixed representations
2- explicit construction of bases suitable for SoV characterization of the spectrum
3- construction of the $Q$-operator in terms of fused transfer matrices
Weaknesses
1- little discussion of relation / unique features of the different constructions
2- notation sometimes confusing
3- English grammar could be improved
Report
Building on their previous work on the complete characterization of the spectrum of integrable quantum chains built on fundamental representations of the Yangians $Y(gl_n)$ and deformations thereof within the separation of variables (SoV) approach to the rational $Y(gl_2)$ spin chains with mixed local higher spin representations and quasiperiodic twisted boundary conditions.
They begin by recalling the construction of these models within the quantum inverse scattering method, in particular the fusion hierarchy of transfer matrices from monodromy matrices with different representations in auxiliary space.
In the following they present three ways to construct a basis of states suitable for the SoV.
(1) following Sklyanin's approach they obtain the quantum separate variables and introduce a covector eigenbasis of the $B$-operator from the monodromy matrix with two-dimensional auxiliary space. The action of the $A$ and $D$ operators on these basis states is given explicitly. Although this not explicitly stated, this allows to formulate the spectral problem in terms of a separable set of linear equations on a lattice with $\text{dim}\mathcal{H}$ sites -- as usual for the SoV in Sklyanin's formulation.
Using the methods developed in their earlier papers the authors construct two additional SoV bases:
(2) by action on a generic covector with the 'fundamental transfer matrices' obtained from monodromy matrices with auxiliary space isomorphic to one of the local quantum spaces. Here the spectrum of the transfer matrix is determined by a system of polynomial equations for $N$ parameters which allows to prove completeness. In fact, these equations are equivalent to the formulation of the spectral problem obtained in (1).
(3) a third 'natural' SoV basis is obtained by action on a generic covector $\langle S|$ with another combination of fused transfer matrices. For a particular choice of $\langle S|$ it coincides with the basis obtained in Sklyanin's approach (1). In this formulation the action of the transfer matrix is straightforward to compute giving the same formulation of the spectral problem as in (1).
Finally, the authors reformulate the spectral problem in terms of a 'quantum spectral curve' (i.e. Baxter type $T$-$Q$ equation) with polynomial $Q$ and reconstruct the $Q$-operator in terms of elements of the monodromy matrix. This allows to generate a SoV basis using the $Q$-operators. Again, a particular choice of a starting covector allows to reproduce the basis (1).
In summary, the authors provide interesting results which have the potential to extend the scope of the SoV approach to the investigation of integrable quantum chains.
Parts of the paper could be more accessible though if the authors would provide more links between the different approaches introduced. Furthermore, the use of notation without introducing it before or never using it again (as with the functions $\alpha$ and $\beta$ in (5.3)) can be obfuscating.
Requested changes
1- it should be mentioned that $P_{ab}$ in (4.4) is
the permutation operator on the spaces involved.
2- Eq. (4.3) does not make sense
3- the role of Proposition 4.2 in Section 4.1 is unclear. If
needed at all, it would be better placed in Section 2.
4- the relation of the states $\langle 0|$ (3.7) and $\langle S|$ (4.20)
used in the constructions (1) and (3) of the SoV basis should be discussed.

---

## Round 2 · Author Response

Dear Editor,

We first would like to thank the referees for their careful reading of the manuscript, for their clarification requests and for pointing out numerous misprints and suggesting several improvements in the notations and also concerning the English grammar. We have implemented the corresponding modifications in our manuscript to take them into account, the mains changes being in Section 2 and in Section 4. We also added a new section, “Conclusion”, and an appendix on the determinant of tri-diagonal matrices that we use for obtaining new fusion relations that enable us to construct in two different ways our SoV bases.

To list the main modifications done in the manuscript (see below), we have answered separately the points raised by each of the two referees.

Best regards,

J. M. Maillet and G. Niccoli

List of the main modifications done in the manuscript:

Answer to Referee 2:

We agree with all remarks of the referee and we have implemented all his requirements.

  1. We have stated that P_{a,b} is the permutation operator.

  2. We have added the missing ^{h_n} in equation (4.3).

  3. As suggested, we have moved the former Proposition 4.2 in the current Proposition 2.3 of Section 2. We kept this proposition as we use it to prove our discrete characterization of the transfer matrix spectrum given in Theorem 4.1.

  4. In our new version of Theorem 4.2 we have presented the construction of the SoV basis both starting from the co-vector <S| (of maximal values of the coordinates h_i = 2 s_i, for each i) and from the co-vector <O| (with coordinates h_i=0 for all I’s), which coincides with the co-vector <0| used in the first Sklyanin’s construction in the case a) of this Theorem. This should clarify the different possible constructions of this SoV basis as required by the referee. The construction of this SoV basis starting from <O| has required the introduction of some additional set of fusion relations introduced in our new Proposition 2.4.

Finally, we have added a section “Conclusion” where we discuss on general grounds the relations between the different SoV bases introduced in the paper. We hope it will enable the reader to have a broader view of the possibilities opened by our new approach to SoV bases.

Answer to Referee 1:

We have implemented mainly all the suggestions and requests of the referee. Then, in the following, we only comment on the few that we didn’t implement completely, or which require some additional explanations.

About 13- We have changed the notation for the quantum determinant.

About 20- We have removed formula (2.30) and explained it by words after current Proposition 2.3 (note that we also enlarged that property to give all central zeroes of the higher fused transfer matrices).

About 24- We have added the missing -1 and introduced in (3.5) the missing definition of the \mathcal{W}.

About 26- We prefer to keep the notation {Sk} instead of using . }

About 33- We would like to keep (4.7) in the current form.

About 36- The difference between the D_{t,n} with or without the superscript is now explained in remark 4.

About 37- The current (4.8) (old (4.10)) is the discrete system of equations completely characterizing the spectrum of the transfer matrix in the SoV basis. We prove that it is equivalent to the characterization in terms of a Baxter’s type TQ-equation which in turn leads to Bethe equations. This is a link with the ABA, however, we would like to stress that we have not done any Ansatz in deriving them. About what is simpler to solve one should say that to our knowledge there do not exist general tools to solve general system of polynomial equations with degree higher than one. Then, one should look to the specific system that we are considering determining if some simplifications emerge (as it should be, due to integrability); this is an open and interesting question. Instead, we have been interested in the reformulation of these polynomial equations in terms of a functional equation (spectral curve equation) leading in its turn to Bethe equations; this is motivated by the fact that for Bethe equations the existing literature is more developed, in particular concerning the analysis of the thermodynamic limit.

About 39- We have defined what common component means and also restated more precisely the use of the Theorem of Bez’out.

About 43- We think that it is appropriate to keep our notation to properly implement the induction as we are proposing it. The N-tuple of \bar h is required to define the level up to which for all the h_i, bigger or equal of the corresponding \bar h_i, we assume that our statement is already satisfied. Now to be able to prove the induction, we have to show that we can move this level reducing of one unit each \bar h_n. To prove it we need that all the h_i for i different of n must be in generic value bigger or equal of the corresponding \bar h_i, we should not just prove it keeping the others h_i on the level \bar h_i. Concerning, the \hat h_n this has been used only to make explicit that we are fixing the value of the h_n to 2s_n.

About 54- We have reformulated the Corollary 5.2 to take into account the suggestion of the referee.

About 53- The condition to be satisfied by any proper “reference co-vector” starting from which an SoV basis can be constructed is the condition (5.46). Now, the “reference co-vector” <L| is written in terms of <S| by an invertible charge in (5.47), so the condition required on <L| is satisfied if <S| satisfies (5.46). In our new section “Conclusion”, we have added a related discussion. In particular, we have shown that a change of “reference co-vector” must always be implemented by the action of an invertible charge on a given “reference co-vector” which is known to satisfy (5.46). So that for the two SoV bases constructions, presented in this paper, this condition explicitly reads in (6.7) and in (6.8).

About 56- We would like to point out that the discrete spectrum characterization of the original transfer matrix, given in Theorem 4.1, defines a simultaneous characterization of the spectrum of all the fused transfer matrices and so also of the fundamental transfer matrix. Indeed, from the simplicity of the transfer matrix spectrum, for any solution of the system (4.8) we get that the formula (2.26) gives also the eigenvalues of the fused transfer matrices. This is done simply by replacing the original transfer matrix in (2.26) by its eigenvalue. The referee probably is asking about the possibility to derive directly a discrete system of equations for the fundamental transfer matrix spectrum. From our previous discussion this system must be equivalent to the one already derived in Theorem 4.1. It might be however interesting to derive it directly, and this should be done using the fusion relations starting from the fundamental transfer matrix and not from the original one.

---

## Round 2 · List of Changes

To list the main modifications done in the manuscript, let us answer separately the points raised by the two referees.

Answer to Referee 2:

We agree with all remarks of the referee and we have implemented all his requirements in the new version of our manuscript.

  1. We have stated that P_{a,b} is the permutation operator.

  2. We have added the missing ^{h_n} in equation (4.3).

  3. As suggested, we have moved the former Proposition 4.2 in the current Proposition 2.3 of Section 2. We kept this proposition as we use it to prove our discrete characterization of the transfer matrix spectrum given in Theorem 4.1.

  4. In our new version of Theorem 4.2 we have presented the construction of the SoV basis both starting from the co-vector <S| (of maximal values of the coordinates h_i = 2 s_i, for each i) and from the co-vector <O| (with coordinates h_i=0 for all i’s), which coincides with the co-vector <0| used in the first Sklyanin’s construction in the case a) of this Theorem. This should clarify the different possible constructions of this SoV basis as required by the referee. The construction of this SoV basis starting from <O| has required the introduction of some additional set of fusion relations introduced in our new Proposition 2.4.

Finally, we have added a section “Conclusion” where we discuss on general grounds the relations between the different SoV bases introduced in the paper. We hope it will also enable the reader to have a broader view of the possibilities opened by our new approach to SoV bases.

Answer to Referee 1:

We have implemented mainly all the suggestions and requests of the referee. Then, in the following, we only comment on the few that we didn’t implement completely, or which require some additional explanations.

About 13- We have changed the notation for the quantum determinant.

About 20- We have removed formula (2.30) and explained it by words after current Proposition 2.3 (note that we also enlarged that property to give all central zeroes of the higher fused transfer matrices).

About 24- We have added the missing -1 and introduced in (3.5) the missing definition of the \mathcal{W}.

About 26- We prefer to keep the notation {Sk} instead of using . }

About 33- We would like to keep (4.7) in the current form.

About 36- The difference between the D_{t,n} with or without the superscript is now explained in remark 4.

About 37- The current (4.8) (old (4.10)) is the discrete system of equations completely characterizing the spectrum of the transfer matrix in the SoV basis. We prove that it is equivalent to the characterization in terms of a Baxter’s type TQ-equation which in turn leads to Bethe equations. This is a link with the ABA, however, we would like to stress that we have not done any Ansatz in deriving them. About what is simpler to solve one should say that to our knowledge there do not exist general tools to solve general system of polynomial equations with degree higher than one. Then, one should look to the specific system that we are considering determining if some simplifications emerge (as it should be, due to integrability); this is an open and interesting question. Instead, we have been interested in the reformulation of these polynomial equations in terms of a functional equation (spectral curve equation) leading in its turn to Bethe equations; this is motivated by the fact that for Bethe equations the existing literature is more developed, in particular concerning the analysis of the thermodynamic limit.

About 39- We have defined what common component means and also restated more precisely the use of the Theorem of Bez’out.

About 43- We think that it is appropriate to keep our notation to properly implement the induction as we are proposing it. The N-tuple of \bar h is required to define the level up to which for all the h_i, bigger or equal of the corresponding \bar h_i, we assume that our statement is already satisfied. Now to be able to prove the induction, we have to show that we can move this level reducing of one unit each \bar h_n. To prove it we need that all the h_i for i different of n must be in generic value bigger or equal of the corresponding \bar h_i, we should not just prove it keeping the others h_i on the level \bar h_i. Concerning, the \hat h_n this has been used only to make explicit that we are fixing the value of the h_n to 2s_n.

About 54- We have reformulated the Corollary 5.2 to take into account the suggestion of the referee.

About 53- The condition to be satisfied by any proper “reference co-vector” starting from which an SoV basis can be constructed is the condition (5.46). Now, the “reference co-vector” <L| is written in terms of <S| by an invertible charge in (5.47), so the condition required on <L| is satisfied if <S| satisfies (5.46). In our new section “Conclusion”, we have added a related discussion. In particular, we have shown that a change of “reference co-vector” must always be implemented by the action of an invertible charge on a given “reference co-vector” which is known to satisfy (5.46). So that for the two SoV bases constructions, presented in this paper, this condition explicitly reads in (6.7) and in (6.8).

About 56- We would like to point out that the discrete spectrum characterization of the original transfer matrix, given in Theorem 4.1, defines a simultaneous characterization of the spectrum of all the fused transfer matrices and so also of the fundamental transfer matrix. Indeed, from the simplicity of the transfer matrix spectrum, for any solution of the system (4.8) we get that the formula (2.26) gives also the eigenvalues of the fused transfer matrices. This is done simply by replacing the original transfer matrix in (2.26) by its eigenvalue. The referee probably is asking about the possibility to derive directly a discrete system of equations for the fundamental transfer matrix spectrum. From our previous discussion this system must be equivalent to the one already derived in Theorem 4.1. It might be however interesting to derive it directly, and this should be done using the fusion relations starting from the fundamental transfer matrix and not from the original one.

---

## Editorial Decision

published